# Above Cloud Aerosol Detection and Retrieval from Multi-Angular Polarimetric Satellite Measurements in a Neural Network Ensemble Approach

Zihao Yuan<sup>1,2</sup>, Guangliang Fu<sup>1</sup>, Hai Xiang Lin<sup>2,3</sup>, Jan Willem Erisman<sup>2</sup>, and Otto P. Hasekamp<sup>1</sup>

Correspondence: Z.Yuan (z.yuan@sron.nl)

**Abstract.** This paper describes an algorithm for above-cloud aerosol (ACA) retrievals from PARASOL (Polarisation and Anisotropy of Reflectances for Atmospheric Science coupled with Observations from a Lidar) Multi-Angle Polarimetric measurements. The algorithm, based on neural networks (NNs), has been trained on synthetic measurements and has been applied to the processing of one-year PARASOL data. The algorithm makes use of three subsequent NNs: 1) for the detection of liquid clouds, 2) for the retrieval of aerosol properties for ACA cases, and 3) an NN forward model to evaluate the goodness-of-fit of the retrieval. The NN's theoretical capability of retrieval is investigated by several synthetic data studies. It is shown that the NNs retrieve ACAOT $_{550}$  (above cloud aerosol optical thickness, at 550 nm), AE $_{440-670}$  (Ångström exponent, between 440 nm and 670 nm), and SSA $_{550}$  (single scattering albedo, at 550 nm) with an RMSE (root mean squared error) of  $\sim 0.1$  on ACAOT $_{550}$ ,  $\sim 0.4$  on AE $_{440-670}$  and  $\sim 0.04$  on SSA $_{550}$  in synthetic experiments. Finally, comparison between the NN retrievals and adjacent PARASOL-RemoTAP clear-sky retrieval in 2008 shows good agreement within the range expected from the synthetic study.

### 1 Introduction

Knowledge about above-cloud aerosol (ACA) is important for understanding aerosol's impact on Earth's energy balance and climate dynamics (Li et al., 2022). From a perspective of aerosol-radiation interaction, it leads to large regional variations in the aerosol direct radiative effect (DRE; Lacagnina et al. (2017); de Graaf et al. (2020); Wilcox (2012)). The sign of the ACA's DRE may differ from that of a clear-sky situation (de Graaf et al., 2023), which depends on a number of factors including the cloud albedo, the aerosol type and its level of absorption (Lenoble et al., 1982; Keil and Haywood, 2003; Peers et al., 2015; Kacenelenbogen et al., 2019). Furthermore, when absorbing aerosols are located above stratocumulus clouds, warming of the layers above the clouds stabilizes the boundary layer, reducing entrainment rates and fostering a moister boundary layer. This may ultimately result in an increased liquid water content and the preservation of cloud cover (Johnson et al., 2004; Brioude et al., 2009). However, uncertainties arise when aerosol and cloud properties are not adequately known, impacting ACA's DRE estimation (de Graaf et al., 2020) and our understanding of aerosol-cloud interaction (Arola et al., 2022). Therefore, obtaining

<sup>&</sup>lt;sup>1</sup>SRON Netherlands Institute for Space Research (NWO-I/SRON), Leiden, the Netherlands

<sup>&</sup>lt;sup>2</sup>Institute of Environmental Science (CML), Leiden University, Leiden, the Netherlands

<sup>&</sup>lt;sup>3</sup>Delft Institute of Applied Mathematics, Delft University of Technology, Delft, the Netherlands

better-retrieved properties for aerosols and clouds in ACA scenarios is important for a comprehensive understanding of the ACA's effect on both radiation and clouds.

25

Satellite-based remote sensing plays a crucial role in quantifying the aerosol direct effect (Myhre et al., 2009; Lacagnina et al., 2015, 2017; Chen et al., 2022) and indirect effect (Gryspeerdt et al., 2017; Hasekamp et al., 2019b; Quaas et al., 2020; Gryspeerdt et al., 2023; Rosenfeld et al., 2024; Jia et al., 2024). For passive sensors, the largest information content on aerosols is available from multi-angle, multiwavelength measurements of both radiance and polarization (Mishchenko and Travis, 1997; Hasekamp and Landgraf, 2007; Dubovik et al., 2019). This type of instrument is referred to as a Multi-Angle Polarimeter (MAP) in this study. Three versions of the Polarization and Directionality of Earth Reflectances (Polder) instrument have flown since 1995. Only Polder-3 on Parasol has provided a multi-year data set between 2004 to 2013. The instrument 3MI (Fougnie et al., 2018), which is an improved version of Polder, is scheduled to launch in 2025 on the Metop SG-A satellite. The NASA PACE mission (Werdell et al., 2019), which launched in February 2024, significantly improves aerosol and cloud retrieval capabilities through advanced MAP measurements, in terms of accuracy as well as spectral and angular sampling. PACE includes two polarimeters: SPEXone (Hasekamp et al., 2019a; Fu et al., 2025), providing hyperspectral measurements at five viewing angles, and HARP-2, providing hyper-angular measurements at four discrete spectral bands. PACE is the first mission in over a decade to deliver advanced MAP data products for aerosols and clouds.

Currently, measurements from satellite-borne MAP instruments can be used to retrieve ACA properties, as the ACA can significantly affect the reflected polarized radiance in a certain range of scattering angles (Knobelspiesse et al., 2015). Initially, Waquet et al. (2009, 2013a) developed a method that retrieves above-cloud aerosol optical thickness (ACAOT) and Ångström exponent (AE) exclusively from polarization measurements. This was achieved using a look-up table (LUT) approach combined with a decision tree strategy. The method was then improved by including additional total radiance measurements (Peers et al., 2015) to simultaneously retrieve the ACA single scattering albedo and the cloud optical thickness (COT) of the cloud layer. Besides MAP instruments, several ACA characterization approaches have been developed for passive and active remote sensing instrument like CALIOP (Cloud-Aerosol Lidar with Orthogonal Polarization), MODIS (Moderate Resolution Imaging Spectroradiometer) and OMI (Ozone Monitoring Instrument). The CALIOP sensor provides high-resolution vertical profiles of aerosols and clouds by measuring attenuated backscatter at 532 and 1064 nm, where extinction and aerosol optical thickness (AOT) is derived from, and depolarization at 532 nm, which helps distinguish particle shape, aiding aerosol classification (Winker et al., 2010; Omar et al., 2009; Hunt et al., 2009). For MODIS and OMI, the "color ratio" method, which utilizes the ratio between the measurements at a shorter (470 nm) and a longer (860 nm) wavelength, is applied to separate AOT from COT (Torres et al., 2012; Jethva et al., 2013). Several studies have shown inter-comparisons between the above data products (Jethva et al., 2014; Deaconu et al., 2017).

The use of Neural Networks (NNs) provides a promising alternative for physics-based and LUT retrievals because of the efficiency in computation and the possibility to take into account the effect on the measured signal of different parameters (e.g., surface reflection) without explicitly retrieving them (e.g., Yuan et al. (2024)). NNs have been used successfully in polarimetric remote sensing of aerosols by e.g. Di Noia et al. (2017), Gao et al. (2021a), Segal-Rozenhaimer et al. (2018), and Gao et al. (2021b), as well as for polarimetric remote sensing of cloud microphysical properties by Di Noia et al. (2019). This work

aims at developing an ACA detection and retrieval scheme for MAP instruments, and focuses on the POLDER-3/PARASOL instrument (hereafter simply referred to as PARASOL) because it is the only MAP with a long-term data set.

The paper is organized as follows: Section 2 introduces the data used in the study, Section 3 describes the NN configurations and the datasets used for the training, Section 4 investigates the performance of the NN on different synthetic datasets, Section 5 shows the data processing of one year (2008) PARASOL measurements and comparison with adjacent PARASOL-RemoTAP clear-sky aerosol retrievals. Finally, Section 6 summarizes and concludes this paper.

### 2 Data description

### 65 2.1 PARASOL

PARASOL (Fougnie et al., 2007) provided multi-angle observations (up to 16 viewing angles per ground pixel) in 9 spectral bands (443, 490, 565, 670, 763, 765, 865, 910, 1020 nm) for intensity and 3 spectral bands for Stokes parameters Q and U (490, 670, 865 nm). The mission was operational in the period 2004-2013 (until 2009 as part of the NASA A-Train satellite constellation). The level 1 data are provided on  $\sim 6 \times 6 \text{ km}^2$  sinusoidally grid. This study uses PARASOL measurements from 6 spectral bands (443, 490, 565, 670, 865, 1020 nm) within latitude ranges from 60° S to 60° N and with at least 14 available viewing angles, as the majority of PARASOL observations contain exactly 14 angles. For measurements with more than 14 available angles, a subset of 14 is selected.

### 2.2 PARASOL RemoTAP aerosol retrievals

In this study, PARASOL RemoTAP (Remote Sensing of Trace Gas and Aerosol Products) aerosol retrievals provide some of the aerosol and surface properties in the training set and are also used for evaluation of the NN ACA retrievals on real PARASOL measurements. The RemoTAP PARASOL retrievals herein (Hasekamp et al., 2024) are based on a parametric 3-mode aerosol description characterized by three log-normal size distribution modes ( $N_{\text{modes}} = 3$ ): one fine mode and two coarse modes (dust and soluble). A detailed overview of RemoTAP can be found in Hasekamp et al. (2024) and Lu et al. (2022).

# 2.3 Cloud phase from MODIS-Aqua cloud product

The MODIS cloud phase product used in this work is generated at 1-km (at nadir) spatial resolutions from MODIS-Aqua L2 data product (MYD06\_L2, Platnick et al. (2015)). Five different cloud flags are categorized in the product: liquid cloud, ice cloud, mixed cloud, uncertain and clear. In this work, a pixel is marked as liquid phase cloud only when the fraction of liquid-cloud-flagged 1-km-resolution MODIS pixels within a 6 km× 6 km PARASOL grid cell is larger than 80%.

### 2.4 AERO-AC above cloud aerosol retrievals

AERO-AC (Waquet et al., 2020) is a global ACA data product from PARASOL measurement, and it is used to compare with the PARASOL-NN ACA retrievals in this paper. In AERO-AC, the ACA properties are only retrieved in case of homogeneous

optically thick (COT > 3) liquid water clouds. The algorithm proceeds to search for the best-fitting aerosol model among all available models, including six fine modes plus a bimodal non-spherical mineral dust particle model. Pixels with partial cloud coverage and cloud edges are removed. Cirrus above liquid water clouds are also filtered and different quality criteria are applied to improve the products.

# 3 Methodology

# 3.1 General settings of the forward simulation

The NN training in this study utilizes synthetic measurements of top-of-atmosphere radiance and degree of linear polarization (DoLP), as a function of wavelength and viewing-solar geometries. The synthetic measurements are generated by the RemoTAP forward model (Hasekamp and Landgraf, 2002, 2005; Schepers et al., 2014), which is a linearized radiative transfer model employed in the RemoTAP retrieval algorithm (Hasekamp et al., 2011; Fu and Hasekamp, 2018; Fu et al., 2020; Lu et al., 2022; Fu et al., 2025). In the calculation of the synthetic measurements, liquid clouds are represented by spherical particles with a Gamma size distribution (Hansen and Travis, 1974), and the refractive index of water is taken from Hess et al. (1998). For ice clouds, hexagonal crystals with varying aspect ratios and surface distortions are used as proxies for variable-complex-shaped ice crystals (van Diedenhoven et al., 2020). The aerosol size distribution follows three log-normal modes, as described in Lu et al. (2022), where each mode is described by the effective radius ( $r_{\rm eff}$ ), effective variance ( $v_{\rm eff}$ ), complex refractive index (dependent on wavelength), AOT<sub>550</sub>, fraction of spherical particles ( $f_{\rm sph}$ ) and aerosol layer height (the central altitude of the Gaussian distributed aerosol profile, FWHM, full width at half maximum, fixed at 2000 m). Here we should note that the forward simulation of ACA scenes includes only fine and dust mode aerosols, while the simulation of clear-sky scenes considers also a soluble coarse mode. The spectrally dependent refractive index  $m(\lambda)$  per mode is parameterized by

$$m(\lambda) = \sum_{k=1}^{n_{\alpha}} \alpha_k m^k(\lambda),\tag{1}$$

where  $m^k(\lambda)$  are prescribed functions of wavelength, for which we use standard refractive index spectra for different aerosol components, i.e., dust (Torres et al., 2007), water-soluble, black carbon (d'Almeida et al., 1991), and organic carbon (Kirchstetter et al., 2004). The Mie- and T-matrix-improved geometrical optics database (Dubovik et al., 2006) is used for the computation from aerosol microphysical properties to optical properties. The ocean reflection properties are parameterized based on wind speed as described in Cox and Munk (1954), and chlorophyll-a concentration as outlined in Fan et al. (2019). For land surface simulations, the bidirectional reflectance distribution function (BRDF) is parameterized using the Ross-Li model (Wanner et al., 1995), while the bidirectional polarization distribution function (BPDF) is parameterized as in Maignan et al. (2009).

The surface (land and ocean) properties for the NN training are from randomly picked pixels of RemoTAP global retrieval for the year 2008. The cloud properties are generated randomly. The aerosol properties are randomly generated values or randomly picked from RemoTAP global retrieval in 2008, depending on the task of different NNs (the details are in Appendix A1, A2 and A3). The geometry combination (solar zenith angle, SZA, viewing zenith angle, VZA and relative azimuth angle,

RAA) are randomly picked from real PARASOL solar-viewing geometries in 2008. Only the measurements with a minimum of 14 angles are considered (see above) for the NN training, in order to evade from a variable-sized input vector to the NN or, as an alternative, an input vector with missing data.

### 3.2 Neural Network training

This work focuses on retrieving the properties of aerosols which are located above a liquid cloud layer, and the retrieval process is depicted in Figure 1. Three NNs are used in the process: 1) NN cloud mask, to select pixels covered by a liquid cloud, 2) NN for aerosol retrieval and 3) NN surrogate radiative transfer model (hereafter referred to as NN forward model). The NN forward model is used to efficiently compute the goodness-of-fit at low computational cost, which is essential for identifying cases where the 1D radiative transfer model breaks down—particularly in scenes with low cloud heterogeneity. Under such conditions, the plane-parallel assumption introduces a positive bias in ACAOT retrievals due to errors in polarized radiance modeling in the cloud bow region (Cornet et al., 2018). These angular inconsistencies are revealed through discrepancies in the fit between forward model and real measurements (Stap et al., 2015, 2016). Additionally, MODIS cloud phase flags are used to mask cases with thin cirrus above liquid cloud (see above).

The first NN (cloud mask) takes intensity, DoLP, and viewing geometries (SZA, VZA, RAA and scattering angle) as input and outputs liquid cloud fraction and ice cloud fraction separately. The independent pixel approximation (IPA) is used to generate partly cloudy cases in the training set, as described in Yuan et al. (2024). The training set consists of 8 million samples including 20% cloud-free pixels, 10% fully covered by liquid cloud, 10% fully covered by ice cloud, and the other 60% partly covered by a mix of liquid cloud and ice cloud. The total cloud fraction is uniformly distributed in a square space (probability density function:  $f(x) = x^2$ ) with more cloud fractions close to 1. This setting reduces the cloud mask's ability when CF< 0.8 but makes it more sensitive at almost fully cloudy cases (cases of interest). The radiative contribution of aerosol and surface properties is also taken into account, as described by Yuan et al. (2024). In the training set of this cloud mask NN, 20% of the samples represent the situation where the aerosol layer is located above the cloud top, in order to improve NN's ability to produce liquid and ice cloud fractions in areas of interest for this study. A pixel will be further processed, if this NN outputs a liquid cloud fraction > 0.8 and an ice cloud fraction 

Figure 1. A flowchart of the NN ACA retrieval process. Three NNs (in the purple rectangles) are used in the process: NN for cloud mask, NN for ACA retrievals and NN forward model. MODIS cloud phase data are used to screen out the residual thin cirrus above liquid clouds. The aerosol retrieval will be discarded if any of the following situations happen: 1) NN liquid cloud fraction < 0.8, 2) NN ice cloud fraction > 0.2, 3) MODIS suggests the cloud phase is not liquid, or 4) the goodness-of-fit ( $\chi^2$ ) > 5.

vector are COT, cloud layer height (CLH), and the liquid droplet  $r_{\rm eff}$  and  $v_{\rm eff}$ . To better represent the real situations, the fine-mode fraction (fraction of fine mode AOT $_{550}$  over the total AOT $_{550}$ ) is randomly taken from PARASOL-RemoTAP clear-sky retrievals, while the total ACAOT $_{550}$  is randomly generated by a log-uniform distribution between 0 and 2. It should be noted that the coarse soluble mode is not considered in this step as it is usually below the cloud layer. An overview of the distribution for the different state vector elements of the training set are given in Table A2. The intensity and DoLP, as a function of wavelength and viewing angle, are compressed using a principal component analysis (PCA) before the training. A total of 25 principal components are retained for radiance (which contains 99.99% explained variance) and 33 for DoLP (which comprise 99.14% explained variance). Different from the training set of the cloud mask NN, the training set of aerosol retrieval NN only contains ACA samples.

The NN for forward calculation is designed to reproduce the MAP measurements from the viewing geometries and the retrieved properties, including aerosol properties of both fine mode and coarse mode and the liquid cloud properties. To make the forward model flexible in viewing geometries, it is trained separately per viewing direction and with the uniformly random-generated SZA, VZA and RAA. For each aerosol retrieval, the NN should be applied 14 times to simulate a MAP measurement at 14 viewing angles. The goodness-of-fit criterion is calculated as:

$$\chi^2 = \frac{1}{n} \sum_{i}^{n} \frac{(\mathbf{y}_i - \mathbf{F}_i)^2}{\sigma_i^2},\tag{2}$$

where n is the total channel of measurements, and  $\mathbf{y}_i$ ,  $\mathbf{F}_i$  respectively stands for the satellite measurements and the NN reproduced measurements at the i-th channel. For the PARASOL measurements in this study, a total of 126 channels are used including 6 wavelengths for intensity and 3 wavelengths for DoLP with 14 viewing angles per wavelength. The noise  $\sigma_i$  is the estimated absolute noise of each channel. Here we use a relative noise of 0.02 for the intensity and an absolute noise of 0.012 for DoLP.

It should be noted that the NN forward model is not a complete forward model. It only works for pixels fully covered by a liquid cloud without any radiative contribution from the surface and is designed only for the purpose of goodness-of-fit assessment for ACA retrievals. The performance of NN forward model on holdout set is shown in Figure 2. The bias of both intensity and DoLP is close to zero. The rstd (relative standard deviation) of intensity is 0.7% and the std (standard deviation) of DoLP is 0.0025. Both of them are below the instrument measurement noise, which suggests the NN forward model is good enough to replace the full physical model (RemoTAP) in estimating goodness-of-fit.

To increase numerical efficiency and reduce memory usage during the training process, we choose the "neural network ensemble" approach (Hansen and Salamon, 1990; Ganaie et al., 2022). In our approach, the whole training set is equally and randomly divided into several parts (further separated into training set, 90% samples, and holdout set, 10% samples), and an individual NN is trained on each part of the training set. The final output is the average of the outputs from all the ensembles. Here, three ensembles are used for liquid cloud mask NN, 16 ensembles for the aerosol retrieval NN, and six ensembles for the NN forward model. The number and size of ensembles is determined by the performance on synthetic validation sets.

**Figure 2.** Scatter plot of intensity and DoLP at 565 nm from NN forward model. The bias of both intensity and DoLP is close to zero. The rstd of intensity is 0.7% and the std of DoLP is 0.0025. Both of them are below the instrument measurement noise, which suggests the NN forward model is good enough to replace the full physical model (RemoTAP) in estimating goodness-of-fit.

For the cloud mask and retrieval NN, we add measurement noise to the training set as a form of regularization (Bishop, 1995). The measurement noise is modeled as a Gaussian random number with a zero mean and a standard deviation of 1 %–3 % relative noise for intensity and 0.012 absolute noise for DoLP.

In this study, Pytorch (version 1.11.0, https://pytorch.org/, last accessed: 11 October 2021) is used to implement the NNs, which are structured as multi-layer perceptrons (MLPs). The training process employs the backpropagation (BP) algorithm (Rumelhart et al., 1986) and batch training with a batch size of 12,000. The performance of NNs in this paper shows little sensitivity to batch size, so a larger batch size is chosen for larger convergence rate (De et al., 2017). The Adam optimizer (Kingma and Ba, 2014) is used to minimize the root mean square error (RMSE) loss function. The architecture of the NN used in this work consists of three hidden layers. We used the settings ( $\gamma = 0.001, \beta_1 = 0.9, \beta_2 = 0.999, \epsilon = 10^{-8}$ ) suggested by Kingma and Ba (2014), where  $\gamma$  is the initial learning rate. For computational efficiency, ReLU is chosen to be the activation function. The liquid cloud mask NN has 64 neurons in each layer, the aerosol retrieval NN has 128 neurons and the NN forward model has 192 neurons. The detailed statistical distribution of the training sets can be found in the appendix A1, A2 and A3.

# 4 Synthetic experiments

To test the compatibility of the algorithm for different aerosol conditions, we apply the NN to three datasets: 1) based on a uniform distribution of the fine-mode fraction between 0 and 1, as a baseline, 2) fine mode dominated cases only (fine-mode fraction > 0.7), and 3) dust mode dominated cases only (fine-mode fraction < 0.3). Details on the statistical distribution of the datasets can be found at Appendix B1. Figure 3 shows the scatter plot of ACAOT<sub>550</sub>, AE<sub>440-670</sub>, and SSA<sub>550</sub> on the three datasets. The AE is calculated using ACAOT at 440 nm and 670 nm.

The retrievals are filtered by a retrieved liquid cloud fraction > 0.8, ice cloud fraction < 0.2, and the goodness-of-fit  $\chi^2$  of the retrieval < 5, all of which can be obtained from the NN for cloud mask and the NN forward model. For AE<sub>440-670</sub> and SSA<sub>550</sub>, an additional mask of retrieved ACAOT<sub>550</sub> > 0.2 is applied. The RMSE is 0.11 for ACAOT<sub>550</sub>, 0.42 for AE<sub>440-670</sub>, and 0.05 for SSA<sub>550</sub> in the mixed dataset. In the fine-dominated dataset, it is 0.11 for ACAOT, 0.55 for AE, and 0.05 for SSA. For the dust-dominated dataset, the RMSE is 0.12 for ACAOT<sub>550</sub>, 0.40 for AE<sub>440-670</sub>, and 0.03 for SSA<sub>550</sub>. Potentially, the NN could be improved by adding more extreme cases to the training set, and this will be a subject of future work.

Additionally, retrieval simulations have been performed on five fully liquid-cloud-covered datasets to investigate the dependence of the retrieval capability on the optical thickness of the underlying liquid cloud. The datasets have the same set of aerosol, cloud, and surface properties (varied within each dataset) but each set has a different (constant) liquid COT between 3 and 40. Each dataset has a total of 10000 samples for both land and ocean. Details on the statistical distribution of the datasets can be found at Appendix B2.

Figure 4 shows the RMSE (over the 10000 retrievals for each COT value), and fraction of successful retrievals as a function of the liquid COT. The retrievals are masked by a retrieved liquid cloud fraction larger than 0.8, an ice cloud fraction smaller than 0.2 (both of which are from the NN cloud mask at original  $6 \times 6$  km<sup>2</sup> resolution) and a goodness-of-fit mask from the NN forward model. The AE<sub>440-670</sub> and SSA<sub>550</sub> retrievals are additionally masked by ACAOT<sub>550</sub> > 0.2. For samples with COT < 5, 80% ocean pixels and almost all land pixels are screened by the cloud mask and goodness-of-fit mask. When the COT is larger than 10 over ocean and 20 over land, the fraction of successful retrievals (that can pass the cloud mask and goodness-of-fit mask) is larger than 80%. For retrievals over land, we see that the RMSE decreases with increasing COT when COT < 20 and then stays constant. This behavior can be explained by the fact that for COT < 20 the measurement is still affected to some extend by the underlying surface which causes a large RMSE. Over ocean, an opposite effect is observed (except for very small COT), because the contribution from the ocean is relatively small and a smaller COT would enhance the relative contribution of the aerosol signal compared to the cloud signal.

### 5 Application on PARASOL data

# 225 5.1 Comparison between PARASOL-NN above cloud aerosol retrievals and adjacent RemoTAP clear-sky aerosol retrievals

The ACA retrievals are evaluated with nearby RemoTAP clear-sky aerosol retrievals in 2008 (Hasekamp et al., 2024) within the same  $1^{\circ} \times 1^{\circ}$  grid cell. If a grid cell contains at least 3 ACA retrievals and at least 3 clear-sky aerosol retrievals, then the comparisons are made by taking the average of the retrieved aerosol properties for both ACA and clear-sky aerosol retrievals, respectively. Figure 5 shows the PARASOL-NN ACA and RemoTAP clear-sky aerosol retrievals in mid-Africa, 4 Aug 2008. In general, it shows large ACAOT<sub>550</sub> ( $\sim$  1) of strongly-absorbing (SSA<sub>550</sub> < 0.85), fine-mode-dominated aerosols (AE<sub>440-670</sub> > 1.5), which is typical in this region, as is also observed in Waquet et al. (2013a) and Chauvigné et al. (2021). In the PARASOL-NN retrieval, the ACAOT<sub>550</sub> is smaller than the adjacent clear-sky AOT<sub>550</sub>, because part of the aerosols are located below the clouds. The ACA seems to be slightly smaller in size (larger AE<sub>440-670</sub>) and more absorbing (lower SSA<sub>550</sub>) than the nearby

**Figure 3.** Scatter plots of ACAOT $_{550}$  (a, d, g), AE $_{440-670}$  (b, e, h), and SSA $_{550}$  (c, f, i) on three synthetic datasets, where one contains both fine-mode-aerosol-dominated cases and dust-mode-aerosol-dominated cases (a, b, c) while others contain only fine-mode-aerosol-dominated cases (d, e, f) or dust-mode-aerosol-dominated cases (g, h, i). The x-axis is the truth of the property and the y-axis is the NN retrieval. The color of each scatter point stands for the number of retrievals (density) on the point. The mean absolute error (MAE), bias, number of retrievals (npix), correlation coefficient (corr) and coefficient of determination ( $R^2$ ) are also given in the plots.

Figure 4. RMSE (a, b, c) and fraction of successful retrievals (d, e, f) as a function of the liquid COT for ACAOT<sub>550</sub> (a, d), AE<sub>440-670</sub> (b, e) and SSA<sub>550</sub> (c, f). The dashed lines are the result of pixels over ocean and the solid lines are over land. The result are both screened by the cloud mask and the goodness-of-fit  $\chi^2 

**Figure 5.** ACAOT<sub>550</sub> and clear-sky AOT<sub>550</sub> (a, b, c), AE<sub>440-670</sub> (d, e, f) and SSA<sub>550</sub> (g, h, i) in mid-Africa, 4 Aug 2008. The left column (a, d, g) shows both the ACA and the clear-sky aerosol retrievals. The middle column (b, e, h) is the ACA retrievals and the right column (c, f, i) is the clear-sky aerosol retrievals. In this case, the ACA (mostly smoke) has a larger AE<sub>440-670</sub> and smaller SSA<sub>550</sub> than the adjacent clear-sky aerosols (smoke and sea salt).

in the synthetic experiment, but in general, the results suggest that the intrinsic aerosol properties (AE and SSA) are more comparable for ACA and adjacent clear-sky aerosol retrievals than the AOT, although the correlation of SSA<sub>550</sub> is low (0.37). To demonstrate the necessity of the goodness-of-fit mask, the comparison without goodness-of-fit mask is shown in figure 4 of the SI, it can be seen the performance of ACAOT<sub>550</sub>, AE<sub>440-670</sub> and SSA<sub>550</sub> become substantially worse.

**Figure 6.** Comparisons of ACA retrievals and clear-sky aerosol retrievals in the same  $1^{\circ} \times 1^{\circ}$  grid. RMSD of total AOT<sub>550</sub> (panel a) is 0.155, fine mode AOT<sub>550</sub> (panel b) is 0.119, AE<sub>440-670</sub> (panel c) is 0.429 and SSA<sub>550</sub> (panel d) is 0.0586. Generally we see a lower ACAOT<sub>550</sub> than the adjacent clear-sky AOT<sub>550</sub>, as part of aerosols are below the cloud. In some cases there is the ACAOT<sub>550</sub> larger than that in clear-sky, and this may be due to contamination of cirrus. The intrinsic aerosol properties (AE and SSA) are more comparable than the AOT.

## 5.2 Comparison between PARASOL-NN and AERO-AC above cloud aerosol retrievals

255 Figure 7 depicts the comparison of ACAOT<sub>670</sub> and AE between PARASOL-NN and AERO-AC at  $1^{\circ} \times 1^{\circ}$  grid for the whole year 2008. The RMSD on ACAOT is 0.094, which is close to 0.107 from synthetic experiments. However, the correlation coefficient on ACAOT is relatively low ( $\sim 0.5$ ), and especially at large ACAOT<sub>670</sub> values from AERO-AC, PARASOL NN algorithm retrieves much lower values. The RMSD on AE is 0.8, which is much greater than in synthetic experiments ( $\sim$ 0.4) and the comparison to adjacent clear-sky retrievals ( $\sim 0.6$ ). For large AE (> 1.5, as predicted by PARASOL-NN), the two data 260 products agree well, but for smaller AE (< 1.5 predicted by PARASOL-NN) the overall agreement is poor. Specifically, there is a group of pixels where AERO-AC predicts values close to  $\sim 1.7$ ). This group of pixels can be explained by a low ACAOT<sub>865</sub> (< 0.1, retrieved by AERO-AC), where the AERO-AC algorithm assumes only fine-mode aerosols in the retrieval. Panel c of Figure 7 shows a comparison where we filter out cases with AERO-AC ACAOT<sub>865</sub> < 0.1 (in addition to the filter ACAOT<sub>550</sub> <0.2 already applied for both AERO-AC and PARASOL-NN). For this comparison, the RMSD is reduced from 0.8 to 0.5 and the 265 correlation coefficient improved from 0.5 to 0.75. Also, clearly the lower limit of  $\sim$ 0.4 in the AERO-AC AE is visible. Besides the reasons mentioned above, the discrepancy may also be caused by the fact that the AE from PARASOL-NN is calculated between 440 and 670 nm while that from AERO-AC is between 670 and 865 nm. To further interpret the differences between our PARASOL-NN algorithm and AERO-AC, we also compared AERO-AC to nearby RemoTAP clear-sky retrievals (see SI Fig 8). From this comparison it follows that the ACOAT<sub>670</sub> from AERO-AC is in general larger than the nearby clear-sky  $AOT_{670}$ , with some very large  $ACAOT_{670}$  values (>2) when the clear-sky  $AOT_{670}$  is < 0.5. This seems to suggest a tendency in AERO-AC to overestimate ACAOT<sub>670</sub>, given that the ACAOT cannot be larger than the total column AOT. The comparison for the AERO-AC AE to clear-sky retrievals shows a similar pattern as the comparison with the above-cloud AE from the PARASOL-NN, although at larger AE the latter agreement is better than the agreement with clear-sky AE. The relatively large AE differences between AERO-AC and NN ACA retrievals (as well as the large AE differences between AERO-AC and PARASOL-RemoTAP clear-sky retrievals) may be related to differences in aerosol model assumptions, AERO-AC relies more on specific aerosol model assumptions under certain conditions, whereas PARASOL-NN and PARASOL-RemoTAP use the same continuous range of aerosol properties for all retrievals. On the other hand, the PARASOL-NN seems to slightly underestimate AE in fine mode dominated cases, based on the synthetic experiments (Figure 3). Moreover, it should be kept in mind that the different wavelength pairs are used for the AE calculation, which may cause discrepancies in the AE value (see Fig 9 in SI). 280

### 5.3 Distribution of the ACA events' frequency and the ACA properties in 2008

285

Figure 8 shows the global seasonal average of ACAOT $_{550}$  and the number of ACA events in spring (Mar–May), summer (Jun–Aug), autumn (Sep–Nov) and winter (Dec, Jan and Feb) on the  $1^{\circ} \times 1^{\circ}$  grid. The average of ACAOT $_{550}$  is calculated only when at least 25 valid PARASOL retrievals are found in the grid cell. The number of ACA events in a cell is defined as the total number of "good" retrievals where ACAOT is larger than 0.1.

**Figure 7.** Comparison of ACAOT $_{670}$  (panel a) and AE (panel b) between PARASOL-NN and AERO-AC. AE from both PARASOL-NN and AERO-AC is filtered with ACAOT $_{550}$  > 0.2 (ACAOT $_{550}$  from AERO-AC is calculated with unfiltered AE and ACAOT $_{670}$ ). Panel c shows the AE comparison with an additional filter: ACAOT $_{865}$  > 0.1 (from AERO-AC). Note the AE given by PARASOL-NN is between 440 nm and 670 nm while that by AERO-AC is between 670 nm and 865 nm.

The results in Figure 8 agree well with the major ACA regions from previous studies (Waquet et al., 2013b; Jethva et al., 2018), which include: 1) Tropical Southeast Atlantic, primarily consisting of biomass burning aerosols. 2) North Pacific, mainly containing industrial pollutants. 3) "Dust Belt" (5-40°N), where mineral dust particles are commonly detected above clouds in this latitudinal band.

The spatial occurrence of ACA events varies largely among each season. In the western coast of mid-Africa, the ACA events occur more in summer and autumn, while in spring and winter, not many events are observed. In the western coast of North America, although the events are detected for all the seasons, fewer events occurred in autumn and winter compared with the other seasons. The events in southeastern China can also be observed for almost all the seasons with somewhat less events in summer and autumn.

295

300

305

When looking into the global seasonal average of ACAOT<sub>550</sub>, we can find two regions with significantly heavy ACA load: the western coast of mid-Africa (mainly summer and autumn, ACAOT<sub>550</sub> > 0.5), western coast of Morocco in north Africa (during summer, ACAOT<sub>550</sub> > 0.5) and northeastern China (during spring, ACAOT<sub>550</sub>  $\sim$  0.2), and these regions are also observed to have a large number of ACA events. In contrast, for some regions with frequent ACA events, such as the western coast of North America, the seasonal average ACAOT is relatively low (ACAOT  $\sim$  0.1). This agrees well with the analyses by Waquet et al. (2013b) in the same year 2008.

We also investigated the annual average of AE and SSA, as is shown in Figure 9. The AE and SSA are calculated where ACAOT $_{550} > 0.2$ . Compared with ACA events in other areas, events around the western coast of mid-Africa exhibit a different characteristic: aerosols have a high AE (indicating smaller particles) and a low SSA (indicating more absorbing components). The high AE and low SSA is an expected feature of the smoke in mid-Africa (Mallet et al., 2024). We have to remark that our AE in regions between  $45^{\circ} - 60^{\circ}$ N and  $45^{\circ} - 60^{\circ}$ S is  $\sim 0.8$ , which differs largely from  $\sim 1.8$  in Waquet et al. (2013b), despite the good agreement of our above cloud AE with the adjacent clear-sky AE in these latitudes. This is because in regions

Figure 8. Seasonal ACAOT $_{550}$  average (a, c, e, g) on each  $1^{\circ} \times 1^{\circ}$  grid and the total number of ACA events (b, d, f, h) on each grid in 2008. From the top to the bottom row, spring (Mar–May), summer (Jun–Aug), autumn (Sep–Nov) and winter (Dec, Jan and Feb). An ACA event is defined when a PARASOL retrieval has passed the cloud mask and goodness-of-fit mask and produces an ACAOT (at 550 nm) larger than 0.1.

**Figure 9.** Annual  $AE_{440-670}$  (panel a) and  $SSA_{550}$  (panel b) average on each  $1^{\circ} \times 1^{\circ}$  grid in 2008. The AE and SSA are calculated where ACAOT<sub>550</sub> > 0.2. The ACA events in the western coast of mid-Africa have a distinctive feature than others, that the aerosols have a large AE (smaller particles) and a smaller SSA (more absorbing).

between  $45^{\circ}$ – $60^{\circ}$ N and  $45^{\circ}$ – $60^{\circ}$ S, the ACAOT<sub>865</sub> retrieved by the AERO-AC algorithm are likely too low to support reliable aerosol type identification, and only fine-mode ACAOT and AE retrievals are performed.

# 6 Conclusion

This paper presents an NN-based approach to detect and retrieve properties of aerosol located above a uniform liquid cloud layer from multi-angle, multi-wavelength polarimetric measurements. The proposed approach is based on a cascade of three NNs trained on synthetic measurements. Separate NNs have been trained for the subtasks of liquid cloud detection, ACA retrieval, and forward modeling for goodness-of-fit calculation. This approach is designed to perform aerosol retrievals for pixels with large liquid cloud cover (CF> 0.8).

We evaluated the approach on different synthetic datasets. The experiment on three datasets (containing both fine- and dust-mode-dominated aerosol, only fine-mode-dominated aerosol and only dust-mode-dominated aerosol) indicates the NNs have the ability to retrieve AOT and SSA from both fine- and dust-mode-dominated aerosol, as well as mixed scenes with an RMSE between 0.10-0.12 for AOT<sub>550</sub> and 0.03-0.05 for SSA<sub>550</sub>. The NNs are also capable to retrieve AE<sub>440-670</sub> with an accuracy that allows separation between fine-mode and dust dominated cases (with an RMSE between 0.40-0.55). The experiments on synthetic data sets with different liquid cloud optical thickness analyze the theoretical sensitivity of the ACA retrieval. Over land, RMSE decreases as COT increases up to 20, then remains constant, likely due to surface influence at lower COT. Over ocean, RMSE shows the opposite trend (except at very low COT), as the relatively small contribution of the ocean surface makes aerosol signals more prominent compared to cloud signals at low COT.

The NN-based approach has been applied to a year of PARASOL data. The retrieved aerosol properties (AOT $_{550}$ , AE $_{440-670}$ , and SSA $_{550}$ ) are compared with adjacent clear-sky RemoTAP-PARASOL aerosol retrievals in the same  $1^{\circ} \times 1^{\circ}$  grid yielding an RMSD of 0.155 for AOT $_{550}$ , 0.429 for AE $_{440-670}$  and 0.0586 for SSA $_{550}$ . The PARASOL-NN ACA retrievals are also compared with the AERO-AC data product (Waquet et al., 2020) and demonstrate reasonably consistent ACAOT $_{670}$  retrievals throughout 2008 with an RMSE of 0.095. In contrast, AE values differ more significantly (RMSD = 0.8), which might be related

to the fact that AERO-AC is based on a limited number of aerosol models, while the PARASOL NN considers a continuous range of aerosol properties. Particularly in areas where AERO-AC yields  $ACAOT_{865} 
  - Hunt, W. H., Winker, D. M., Vaughan, M. A., Powell, K. A., Lucker, P. L., and Weimer, C.: CALIPSO Lidar Description and Performance Assessment, Journal of Atmospheric and Oceanic Technology, 26, 1214 1228, https://doi.org/10.1175/2009JTECHA1223.1, 2009.

- Jethva, H., Torres, O., Remer, L., and Bhartia, P.: A Color Ratio Method for Simultaneous Retrieval of Aerosol and Cloud Optical Thickness of Above-Cloud Absorbing Aerosols From Passive Sensors: Application to MODIS Measurements, IEEE Transactions on Geoscience and Remote Sensing, 51, https://doi.org/10.1109/TGRS.2012.2230008, 2013.
- Jethva, H., Torres, O., Waquet, F., Chand, D., and Hu, Y.: How do A-train sensors intercompare in the retrieval of above-cloud aerosol optical depth? A case study-based assessment, Geophysical Research Letters, 41, 186–192, https://doi.org/https://doi.org/10.1002/2013GL058405, 2014.
  - Jethva, H., Torres, O., and Ahn, C.: A 12-year long global record of optical depth of absorbing aerosols above the clouds derived from the OMI/OMACA algorithm, Atmospheric Measurement Techniques, 11, 5837–5864, https://doi.org/10.5194/amt-11-5837-2018, 2018.

- Jia, H., Hasekamp, O., and Quaas, J.: Revisiting Aerosol–Cloud Interactions From Weekly Cycles, Geophysical Research Letters, 51, e2024GL108 266, https://doi.org/https://doi.org/10.1029/2024GL108266, e2024GL108266 2024GL108266, 2024.
  - Johnson, B. T., Shine, K. P., and Forster, P. M.: The semi-direct aerosol effect: Impact of absorbing aerosols on marine stratocumulus, Quarterly Journal of the Royal Meteorological Society, 130, 1407–1422, https://doi.org/10.1256/qj.03.61, 2004.
- Kacenelenbogen, M. S., Vaughan, M. A., Redemann, J., Young, S. A., Liu, Z., Hu, Y., Omar, A. H., LeBlanc, S., Shinozuka, Y., Livingston, J., Zhang, Q., and Powell, K. A.: Estimations of global shortwave direct aerosol radiative effects above opaque water clouds using a combination of A-Train satellite sensors, Atmos. Chem. Phys., 19, 4933–4962, https://doi.org/10.5194/acp-19-4933-2019, 2019.
  - Keil, A. and Haywood, J. M.: Solar radiative forcing by biomass burning aerosol particles during SAFARI 2000: A case study based on measured aerosol and cloud properties, Journal of Geophysical Research: Atmospheres, 108, 8467, https://doi.org/10.1029/2002JD002315, 2003.
  - Kingma, D. and Ba, J.: Adam: A Method for Stochastic Optimization, in: International Conference on Learning Representations, 2014.
- Kirchstetter, T. W., Novakov, T., and Hobbs, P. V.: Evidence that the spectral dependence of light absorption by aerosols is affected by organic carbon, Journal of Geophysical Research, 109, D21 208, https://doi.org/10.1029/2004JD004999, 2004.
  - Knobelspiesse, K., Cairns, B., Jethva, H., Kacenelenbogen, M., Segal Rozenhaimer, M., and Torres, O.: Remote sensing of above cloud aerosols, Light Scattering Reviews 9: Light Scattering and Radiative Transfer, pp. 167–210, https://doi.org/10.1007/978-3-642-37985-7 5, 2015.
- 495 Lacagnina, C., Hasekamp, O., Bian, H., Curci, G., Myhre, G., Noije, T., Michael, S., Skeie, R., Takemura, T., and Zhang, K.: Aerosol single-scattering albedo over the global oceans: Comparing PARASOL retrievals with AERONET, OMI, and AeroCom models estimates, Journal of Geophysical Research: Atmospheres, 120, 9814–9836, https://doi.org/10.1002/2015JD023501, 2015.
  - Lacagnina, C., Hasekamp, O. P., and Torres, O.: Direct radiative effect of aerosols based on PARASOL and OMI satellite observations, Journal of Geophysical Research: Atmospheres, 122, 1777–1791, https://doi.org/10.1002/2016JD025706, 2017.
- Lenoble, J., Tanre, D., Deschamps, P. Y., and Herman, M.: A Simple Method to Compute the Change in Earth-Atmosphere Radiative Balance Due to a Stratospheric Aerosol Layer, Journal of Atmospheric Science, 39, 2565–2576, https://doi.org/10.1175/1520-0469(1982)039<2565:ASMTCT>2.0.CO;2, 1982.
  - Li, J., Carlson, B. E., Yung, Y. L., and et al.: Scattering and absorbing aerosols in the climate system, Nat Rev Earth Environ, 3, 363–379, https://doi.org/10.1038/s43017-022-00296-7, 2022.
- Lu, S., Landgraf, J., Fu, G., van Diedenhoven, B., Wu, L., Rusli, S., and Hasekamp, O.: Simultaneous Retrieval of Trace Gases, Aerosols, and Cirrus Using RemoTAP—The Global Orbit Ensemble Study for the CO2M Mission, Frontiers in Remote Sensing, 3, 914378, https://doi.org/10.3389/frsen.2022.914378, 2022.
  - Maignan, F., Breon, F.-M., Fédèle, E., and Bouvier, M.: Polarized reflectances of natural surfaces: Spaceborne measurements and analytical modeling, Remote Sensing of Environment, 113, 2642–2650, https://doi.org/10.1016/j.rse.2009.07.022, 2009.
- Mallet, M., Voldoire, A., Solmon, F., Nabat, P., Drugé, T., and Roehrig, R.: Impact of biomass burning aerosols (BBA) on the tropical African climate in an ocean–atmosphere–aerosol coupled climate model, Atmospheric Chemistry and Physics, 24, 12509–12535, https://doi.org/10.5194/acp-24-12509-2024, 2024.
  - Mishchenko, M. I. and Travis, L. D.: Satellite retrieval of aerosol properties over the ocean using polarization as well as intensity of reflected sunlight, Journal of Geophysical Research: Atmospheres, 102, 16989–17013, https://doi.org/10.1029/96JD02425, 1997.

Myhre, G., Berglen, T. F., Johnsrud, M., Hoyle, C. R., Berntsen, T. K., Christopher, S. A., Fahey, D. W., Isaksen, I. S. A., Jones, T. A., Kahn, R. A., Loeb, N., Quinn, P., Remer, L., Schwarz, J. P., and Yttri, K. E.: Modelled radiative forcing of the direct aerosol effect with multi-observation evaluation, Atmospheric Chemistry and Physics, 9, 1365–1392, https://doi.org/10.5194/acp-9-1365-2009, 2009.

520

540

- Omar, A. H., Winker, D. M., Vaughan, M. A., Hu, Y., Trepte, C. R., Ferrare, R. A., Lee, K.-P., Hostetler, C. A., Kittaka, C., Rogers, R. R., Kuehn, R. E., and Liu, Z.: The CALIPSO Automated Aerosol Classification and Lidar Ratio Selection Algorithm, Journal of Atmospheric and Oceanic Technology, 26, 1994 2014, https://doi.org/10.1175/2009JTECHA1231.1, 2009.
- Peers, F., Waquet, F., Cornet, C., Dubuisson, P., Ducos, F., Goloub, P., Szczap, F., Tanré, D., and Thieuleux, F.: Absorption of aerosols above clouds from POLDER/PARASOL measurements and estimation of their direct radiative effect, Atmospheric Chemistry and Physics, 15, 4179–4196, https://doi.org/10.5194/acp-15-4179-2015, 2015.
- Platnick, S., Ackerman, S., King, M. D., et al.: MODIS Atmosphere L2 Cloud Product (06\_L2), http://dx.doi.org/10.5067/MODIS/MYD06\_ 525 L2.061, 2015.
  - Quaas, J., Arola, A., Cairns, B., Christensen, M., Deneke, H., Ekman, A., Feingold, G., Fridlind, A., Gryspeerdt, E., Hasekamp, O., Li, Z., Lipponen, A., Ma, P.-L., Mülmenstädt, J., Nenes, A., Penner, J., Rosenfeld, D., Schrödner, R., Sinclair, K., and Wendisch, M.: Constraining the Twomey effect from satellite observations: issues and perspectives, Atmospheric Chemistry and Physics, 20, 15 079–15 099, https://doi.org/10.5194/acp-20-15079-2020, 2020.
- Rosenfeld, D., Kokhanovsky, A., Goren, T., Gryspeerdt, E., Hasekamp, O., Jia, H., Lopatin, A., Quaas, J., Pan, Z., and Sourdeval, O.: Challenges in measuring aerosol cloud-mediated radiative forcing, Eos, 105, https://doi.org/10.1029/2024EO245012, published on 29 February 2024, 2024.
  - Rumelhart, D., Hinton, G., and Williams, R.: Learning representations by back-propagating errors, Nature, 323, 533–536, https://doi.org/10.1038/323533a0, 1986.
- Schepers, D., Brugh, J., Hahne, P., Butz, A., Hasekamp, O., and Landgraf, J.: LINTRAN v2.0: A linearised vector radiative transfer model for efficient simulation of satellite-born nadir-viewing reflection measurements of cloudy atmospheres, Journal of Quantitative Spectroscopy and Radiative Transfer, 149, 347–359, https://doi.org/10.1016/j.jqsrt.2014.08.019, 2014.
  - Segal-Rozenhaimer, M., Miller, D. J., Knobelspiesse, K., Redemann, J., Cairns, B., and Alexandrov, M. D.: Development of neural network retrievals of liquid cloud properties from multi-angle polarimetric observations, Journal of Quantitative Spectroscopy and Radiative Transfer, 220, 39–51, https://doi.org/https://doi.org/10.1016/j.jqsrt.2018.08.030, 2018.
  - Spilling, D. and Thales, A.: The Multi Angle Polarimeter (MAP) on board ESA's Copernicus Carbon Dioxide Monitoring mission (CO2M), in: Proc. SPIE 11852, International Conference on Space Optics ICSO 2020, p. 118520R, https://doi.org/10.1117/12.2599174, 2021.
  - Stap, F., Hasekamp, O., Emde, C., and Röckmann, T.: Influence of 3D Effects on 1D Aerosol Retrievals in Synthetic, Partially Clouded Scenes, Journal of Quantitative Spectroscopy and Radiative Transfer, 170, https://doi.org/10.1016/j.jqsrt.2015.10.008, 2015.
- Stap, F., Hasekamp, O., Emde, C., and Röckmann, T.: Multiangle photopolarimetric aerosol retrievals in the vicinity of clouds: Synthetic study based on a large eddy simulation, Journal of Geophysical Research: Atmospheres, 121, https://doi.org/10.1002/2016JD024787, 2016.
  - Torres, O., Tanskanen, A., Veihelmann, B., Ahn, C., Braak, R., Bhartia, P. K., Veefkind, P., and Levelt, P.: Aerosols and surface UV products from Ozone Monitoring Instrument observations: An overview, Journal of Geophysical Research, 112, D24S47, https://doi.org/10.1029/2007JD008809, 2007.
  - Torres, O., Jethva, H., and Bhartia, P. K.: Retrieval of Aerosol Optical Depth above Clouds from OMI Observations: Sensitivity Analysis and Case Studies, J. Atmos. Sci., 69, 1037–1053, https://doi.org/10.1175/JAS-D-11-0130.1, 2012.

- van Diedenhoven, B., Ackerman, A. S., Fridlind, A. M., Cairns, B., and Riedi, J.: Global statistics of ice microphysical and optical properties at tops of optically thick ice clouds, Journal of Geophysical Research: Atmospheres, 125, e2019JD031811, https://doi.org/10.1029/2019JD031811, 2020.
  - Wanner, W., Li, X., and Strahler, A. H.: On the derivation of kernels for kernel-driven models of bidirectional reflectance, Journal of Geophysical Research: Atmospheres, 100, 21 077–21 089, https://doi.org/10.1029/95JD02371, 1995.
  - Waquet, F., Riedi, J., C.-Labonnote, L., Goloub, P., Cairns, B., Deuzé, J.-L., and Tanré, D.: Aerosol Remote Sensing over Clouds Using A-Train Observations, Journal of The Atmospheric Sciences J ATMOS SCI, 66, https://doi.org/10.1175/2009JAS3026.1, 2009.
- Waquet, F., Cornet, C., Deuzé, J.-L., Dubovik, O., Ducos, F., Goloub, P., Herman, M., Lapyonok, T., Labonnote, L. C., Riedi, J., Tanré, D., Thieuleux, F., and Vanbauce, C.: Retrieval of aerosol microphysical and optical properties above liquid clouds from POLDER/PARASOL polarization measurements, Atmospheric Measurement Techniques, 6, 991–1016, https://doi.org/10.5194/amt-6-991-2013, 2013a.
  - Waquet, F., Peers, F., Ducos, F., Goloub, P., Platnick, S., Riedi, J., Tanré, D., and Thieuleux, F.: Global analysis of aerosol properties above clouds, Geophysical Research Letters, 40, 5809–5814, https://doi.org/https://doi.org/10.1002/2013GL057482, 2013b.
- Waquet, F., Peers, F., Ducos, F., Thieuleux, F., Deaconu, L., Chauvigné, A., and Riedi, J.: Aerosols above clouds products from POLDER/-PARASOL satellite observations (AERO-AC products), https://doi.org/10.25326/82, 2020.
  - Werdell, P., Behrenfeld, M., Bontempi, P., Boss, E., Cairns, B., Davis, G., Franz, B., Gliese, U., Gorman, E., Hasekamp, O., Knobelspiesse, K., Mannino, A., Martins, V., Mcclain, C., Meister, G., and Remer, L.: The Plankton, Aerosol, Cloud, ocean Ecosystem (PACE) mission: Status, science, advances, Bulletin of the American Meteorological Society, 100, https://doi.org/10.1175/BAMS-D-18-0056.1, 2019.
- Wilcox, E. M.: Direct and semi-direct radiative forcing of smoke aerosols over clouds, Atmospheric Chemistry and Physics, 12, 139–149, https://doi.org/10.5194/acp-12-139-2012, 2012.
  - Winker, D. M., Pelon, J., Coakley, J. A., Ackerman, S. A., Charlson, R. J., Colarco, P. R., Flamant, P., Fu, Q., Hoff, R. M., Kittaka, C., Kubar, T. L., Treut, H. L., Mccormick, M. P., Mégie, G., Poole, L., Powell, K., Trepte, C., Vaughan, M. A., and Wielicki, B. A.: The CALIPSO Mission: A Global 3D View of Aerosols and Clouds, Bulletin of the American Meteorological Society, 91, 1211 1230, https://doi.org/10.1175/2010BAMS3009.1, 2010.
  - Yuan, Z., Fu, G., van Diedenhoven, B., Lin, H. X., Erisman, J. W., and Hasekamp, O. P.: Cloud detection from multi-angular polarimetric satellite measurements using a neural network ensemble approach, Atmospheric Measurement Techniques, 17, 2595–2610, https://doi.org/10.5194/amt-17-2595-2024, 2024.

### Appendix A: Statistical distributions of the training data sets for the different NNs

**Table A1.** Details of the statistical distributions of the aerosol and cloud parameters used to generate the training datasets for cloud mask NN. Distribution of "RemoTAP" means properties are randomly taken from 2008 PARASOL-RemoTAP L2 database.

| parameter                                   | min   | max   | mean | distribution |
|---------------------------------------------|-------|-------|------|--------------|
| wind speed (m/s)                            | 0.1   | 87    | 7.52 | RemoTAP      |
| $chl-\alpha$ concentration                  | 0.001 | 10    | 1.92 | RemoTAP      |
| Li-sparse                                   | 0     | 0.35  | 0.14 | RemoTAP      |
| Ross-thick                                  | 0     | 1.4   | 0.41 | RemoTAP      |
| Maignan bpdf                                | 0.2   | 10    | 3.02 | RemoTAP      |
| brdf scaling coefficient (443nm)            | 0     | 0.40  | 0.06 | RemoTAP      |
| brdf scaling coefficient (490nm)            | 0     | 0.45  | 0.10 | RemoTAP      |
| brdf scaling coefficient (565nm)            | 0     | 0.50  | 0.17 | RemoTAP      |
| brdf scaling coefficient (670nm)            | 0     | 0.65  | 0.23 | RemoTAP      |
| brdf scaling coefficient (865nm)            | 0     | 0.80  | 0.33 | RemoTAP      |
| brdf scaling coefficient (1020nm)           | 0     | 0.90  | 0.37 | RemoTAP      |
| effective radius of liquid cloud ( $\mu$ m) | 3     | 25    | 14   | uniform      |
| effective variance of liquid cloud          | 0.03  | 0.35  | 0.19 | uniform      |
| cloud optical thickness of liquid cloud     | 1     | 40    | 10.6 | log-uniform  |
| cloud layer height of liquid cloud (km)     | 1     | 8     | 5.5  | uniform      |
| effective radius of ice cloud ( $\mu$ m)    | 10    | 60    | 30   | uniform      |
| cloud optical thickness of ice cloud        | 1     | 100   | 21.5 | log-uniform  |
| cloud layer height of ice cloud (km)        | 8     | 17    | 9.5  | uniform      |
| aspect ratio of ice cloud crystals          | 0.179 | 5.592 | 1.57 | log-uniform  |
| distortion of ice cloud crystals            | 0.1   | 0.7   | 0.4  | uniform      |
| aerosol effective radius of fine mode       | 0.02  | 0.57  | 0.14 | RemoTAP      |
| aerosol effective variance of fine mode     | 0.01  | 0.8   | 0.20 | RemoTAP      |
| aerosol optical thickness of fine mode      | 0     | 4.58  | 0.67 | log-uniform  |
| aerosol effective radius of dust mode       | 0.7   | 6.12  | 1.89 | RemoTAP      |
| aerosol effective variance of dust mode     | 0.01  | 0.8   | 0.58 | RemoTAP      |
| aerosol optical thickness of dust mode      | 0     | 3.95  | 0.60 | log-uniform  |
| aerosol effective radius of soluble mode    | 0.7   | 6.12  | 3.24 | RemoTAP      |
| aerosol effective variance of soluble mode  | 0.01  | 0.8   | 0.59 | RemoTAP      |
| aerosol optical thickness of soluble mode   | 0     | 3.95  | 0.60 | log-uniform  |

**Table A2.** Details of the statistical distributions of the aerosol and cloud parameters used to generate the training datasets for NN ACA retrieval. Distribution of "RemoTAP" means properties are randomly taken from 2008 PARASOL-RemoTAP L2 database.

| parameter                                          | min  | max  | mean | distribution |
|----------------------------------------------------|------|------|------|--------------|
| effective radius of liquid cloud ( $\mu$ m)        | 3    | 25   | 14   | uniform      |
| effective variance of liquid cloud                 | 0.03 | 0.35 | 0.19 | uniform      |
| cloud optical thickness of liquid cloud            | 3    | 40   | 14.3 | log-uniform  |
| cloud layer height of liquid cloud (km)            | 0.4  | 4    | 2.2  | uniform      |
| aerosol effective radius of fine mode              | 0.02 | 0.57 | 0.14 | RemoTAP      |
| aerosol effective variance of fine mode            | 0.01 | 0.8  | 0.20 | RemoTAP      |
| above cloud aerosol optical thickness of fine mode | 0    | 2    | 0.26 | log-uniform  |
| aerosol effective radius of dust mode              | 0.7  | 6.12 | 1.89 | RemoTAP      |
| aerosol effective variance of dust mode            | 0.01 | 0.8  | 0.58 | RemoTAP      |
| above cloud aerosol optical thickness of dust mode | 0    | 2    | 0.26 | log-uniform  |

**Table A3.** Details of the statistical distributions of the aerosol and cloud parameters used to generate the training datasets for NN forward model. The range of aerosol effective radius and effective variance for both fine mode and dust mode is smaller than that for the retrieval, because here it takes no extreme cases into account, which is relatively rare.

| parameter                                          | min  | max  | mean | distribution |
|----------------------------------------------------|------|------|------|--------------|
| effective radius of liquid cloud ( $\mu$ m)        | 3    | 25   | 14   | uniform      |
| effective variance of liquid cloud                 | 0.03 | 0.35 | 0.19 | uniform      |
| cloud optical thickness of liquid cloud            | 3    | 40   | 14.3 | log-uniform  |
| cloud layer height of liquid cloud (km)            | 0.4  | 4    | 2.2  | uniform      |
| aerosol effective radius of fine mode              | 0.03 | 0.3  | 0.15 | uniform      |
| aerosol effective variance of fine mode            | 0.1  | 0.3  | 0.20 | uniform      |
| above cloud aerosol optical thickness of fine mode | 0    | 2    | 0.26 | log-uniform  |
| aerosol effective radius of dust mode              | 0.8  | 3.0  | 1.9  | uniform      |
| aerosol effective variance of dust mode            | 0.4  | 0.8  | 0.6  | uniform      |
| above cloud aerosol optical thickness of dust mode | 0    | 2    | 0.26 | log-uniform  |

# Appendix B: Statistical distributions of the synthetic datasets for testing

**Table B1.** Details of the statistical distributions of the aerosol and cloud parameters used to generate the datasets for experiment of fine mode, dust mode seperate and together. Distribution of "RemoTAP" means properties are randomly taken from 2008 PARASOL-RemoTAP L2 database.

| parameter                                                      | min  | max  | mean | distribution |
|----------------------------------------------------------------|------|------|------|--------------|
| effective radius of liquid cloud ( $\mu$ m)                    | 3    | 25   | 14   | uniform      |
| effective variance of liquid cloud                             | 0.03 | 0.35 | 0.19 | uniform      |
| cloud optical thickness of liquid cloud                        | 3    | 40   | 14.3 | log-uniform  |
| cloud layer height of liquid cloud (km)                        | 0.4  | 4    | 2.2  | uniform      |
| aerosol effective radius of fine mode                          | 0.02 | 0.57 | 0.14 | RemoTAP      |
| aerosol effective variance of fine mode                        | 0.01 | 0.8  | 0.20 | RemoTAP      |
| above cloud aerosol optical thickness of fine mode (if exists) | 0    | 2    | 0.26 | log-uniform  |
| aerosol effective radius of dust mode                          | 0.7  | 6.12 | 1.89 | RemoTAP      |
| aerosol effective variance of dust mode                        | 0.01 | 0.8  | 0.58 | RemoTAP      |
| above cloud aerosol optical thickness of dust mode (if exists) | 0    | 2    | 0.26 | log-uniform  |

**Table B2.** Details of the statistical distributions of the aerosol and cloud parameters used to generate the datasets for sensitivity analysis of underlying liquid cloud optical thickness. Distribution of "RemoTAP" means properties are randomly taken from 2008 PARASOL-RemoTAP L2 database. COT of liquid cloud (distribution "special") is constant in each experiment for sensitivity tests.

| parameter                                          | min   | max  | mean | distribution |
|----------------------------------------------------|-------|------|------|--------------|
| wind speed (m/s)                                   | 0.1   | 87   | 7.52 | RemoTAP      |
| $\mathrm{chl}	ext{-}lpha$ concentration            | 0.001 | 10   | 1.92 | RemoTAP      |
| Li-sparse                                          | 0     | 0.35 | 0.14 | RemoTAP      |
| Ross-thick                                         | 0     | 1.4  | 0.41 | RemoTAP      |
| Maignan bpdf                                       | 0.2   | 10   | 3.02 | RemoTAP      |
| brdf scaling coefficient (443nm)                   | 0     | 0.40 | 0.06 | RemoTAP      |
| brdf scaling coefficient (490nm)                   | 0     | 0.45 | 0.10 | RemoTAP      |
| brdf scaling coefficient (565nm)                   | 0     | 0.50 | 0.17 | RemoTAP      |
| brdf scaling coefficient (670nm)                   | 0     | 0.65 | 0.23 | RemoTAP      |
| brdf scaling coefficient (865nm)                   | 0     | 0.80 | 0.33 | RemoTAP      |
| brdf scaling coefficient (1020nm)                  | 0     | 0.90 | 0.37 | RemoTAP      |
| effective radius of liquid cloud ( $\mu$ m)        | 3     | 25   | 14   | uniform      |
| effective variance of liquid cloud                 | 0.03  | 0.35 | 0.19 | uniform      |
| cloud optical thickness of liquid cloud            | 3     | 40   | N/A  | special      |
| cloud layer height of liquid cloud (km)            | 0.4   | 4    | 2.2  | uniform      |
| aerosol effective radius of fine mode              | 0.02  | 0.57 | 0.14 | RemoTAP      |
| aerosol effective variance of fine mode            | 0.01  | 0.8  | 0.20 | RemoTAP      |
| above cloud aerosol optical thickness of fine mode | 0     | 2    | 0.26 | log-uniform  |
| aerosol effective radius of dust mode              | 0.7   | 6.12 | 1.89 | RemoTAP      |
| aerosol effective variance of dust mode            | 0.01  | 0.8  | 0.58 | RemoTAP      |
| above cloud aerosol optical thickness of dust mode | 0     | 2    | 0.26 | log-uniform  |