# Peer review of "Above Cloud Aerosol Detection and Retrieval from Multi-Angular Polarimetric Satellite Measurements in a Neural Network Ensemble Approach"

_EGUsphere, 2025_

## Referee Comment (RC2)

Review of the article intitled: "Above Cloud Aerosol Detection and Retrieval from Multi-Angular Polarimetric Satellite Measurements in a Neural Network Ensemble Approach" by Yuan et al. for publication in AMT.

Anonymous reviewer.

**Opinion on the publication:**

Major review: Significant corrections, clarifications, and additions are required before this article can be accepted.

**General description:**

This article describes an algorithm for detecting aerosols above clouds using the spectral, angular, and polarized radiance data provided by the POLDER spaceborne instrument. The method is potentially applicable to future other instruments that will provide similar data. They use some of the concept and work previously conducted on aerosol detection above clouds with the POLDER instrument using a LUT-type method, but this time with an AI method, using neural networks.

The article details a three-step methodology for remote sensing of aerosols in cloudy scenes: liquid cloud detection, aerosol property retrieval above clouds, and a posteriori radiance simulation. Details of the datasets employed in training the neural network, specifically regarding aerosol and cloud particle properties, are provided. The method is evaluated using synthetic data, then applied to a year of POLDER instrument data. Aerosol properties are compared to those from an established clear-sky algorithm, though direct comparison is limited as the latter covers the total atmospheric column, not just above clouds. A brief qualitative discussion compares the new method's results with those from POLDER's operational algorithm for aerosols above clouds.

**General review:**

The use of AI methods in aerosol remote sensing, especially for complex cloudy scenes, is a promising and justified approach. While the geophysical results are quite convincing, I have reservations regarding several aspects of the article:

-The description of the neural network's characteristics and the assessment of its performance is too brief.

-The state of the art requires significant enrichment and a thorough refresh.

-The algorithm's results have not been quantitatively validated or compared against equivalent products.

More specifically, the individual networks aren't evaluated; only the overall solution is. Key methodological choices are not justified, and some information on the network structure, are missing, making replication difficult. Correlations between retrieved and test parameters are often quite low, and the paper lacks sufficient metrics in tables for readers to adequately assess the method's accuracy. Furthermore, the AI references are outdated, which is surprising given the rapid advancements in this field.

It's also important to include comprehensive references to existing operational methods for detecting aerosols above clouds, particularly those developed for passive and active instruments. The community has made significant comparisons between these active and passive methods, both for case studies and global applications, and the associated articles should be thoroughly cited. For the POLDER/PARASOL method (Waquet et al., 2013; Peers et al., 2015), a review of its technical aspects and geophysical results is needed. Technical details of this method would better explain the differences between the results presented in this article and the operational POLDER algorithm.

Finally, is seems that the proposed method doesn't account for sub-pixel cloud heterogeneity effects. This could lead to biases, potentially overestimating aerosol optical depth above clouds. At the very last, the limitations of using plane-parallel radiative transfer code for aerosol remote sensing in cloudy scenes should be mentioned.

My detailed comments and questions follow below.

**Abstract.**

Please add the wavelength(s) used for ACAOT, AE and SSA

**Introduction.**

You compare your results with those from the method previously developed for POLDER (Waquet et al., 2013, Peers et al., 2015). The parameters retrieved with your method and many of the underlying concepts, such as using MODIS data to identify and remove cirrus clouds and restricting retrievals to high cloud fraction and large COT values, are similar to those of the previously cited method.

So, this section requires improved referencing and description of the operational aerosol above cloud algorithm previously developed for POLDER/PARASOL, including the associated available product and validation efforts. The AERO-AC product, with its DOI, is globally available for 5 years of POLDER data, which is worth noting for the reader. https://www.icare.univ-lille.fr/aero-ac/

Some of the following explanations should be incorporated into the manuscript (see also my additional comments at the end of this review).

- Initially, the Waquet et al. (2009, 2013) method determined above-cloud aerosol optical thickness and Ångström exponent exclusively from polarization measurements. This was achieved using a look-up table (LUT) approach combined with a decision tree strategy**.**
-The method was then improved by including additional total radiance measurements (Peers et al., 2015) to simultaneously retrieve the above cloud aerosol single scattering albedo and the cloud optical thickness of the below cloud layer (COT).
-The associated global product is referred to as AERO-AC (Waquet et al., 2020)
- The aerosol above cloud properties are only retrieved in case of homogeneous optically thick (COT > 3) and liquid water clouds. Cloud fractional covers and cloud edges are removed. Cirrus above liquid water clouds are also filtered and different quality criteria are eventually applied to improve the products.

Please add the following references:

Peers, F., Waquet, F., Cornet, C., Dubuisson, P., Ducos, F., Goloub, P., Szczap, F., Tanré, D., and Thieuleux, F.: Absorption of aerosols above clouds from POLDER/PARASOL measurements and estimation of their direct radiative effect, Atmos. Chem. Phys., 15, 4179–4196, https://doi.org/10.5194/acp-15-4179-2015, 2015.

Waquet F., Peers F., Ducos F., Thieuleux F., Deaconu L., A. Chauvigné and Riedi, J.: Aerosols above clouds products from POLDER/PARASOL satellite observations (AERO-AC products), doi:10.25326/82, 2020.

Please mention the methods that use active measurements to retrieve aerosol properties above clouds. Different methods (standard methods and advanced methods like the "depolarization ratio method") were developed for CALIOP and various products are available (see Jethva et al., (2014) and Deaconu et al. (2017))

It's also important to highlight the research community's dedication to validating and intercomparing their passive and active aerosol-above-cloud products. This has involved rigorous work, ranging from in-depth case study analyses (Jethva et al., 2014) — supported by airborne sun-photometer data (Chauvigné et al., 2021) — to comprehensive global scale analyses (Deaconu etal., 2017).

Please add the following references:

Jethva, H., O. Torres, F. Waquet, D. Chand, and Y. Hu (2014), How do A-train sensorsintercompare in the retrieval of above-cloud aerosol optical depth? A case study-based assessment, Geophys. Res. Lett., 41, 186–192, doi:10.1002/2013GL058405.

Deaconu, L. T., Waquet, F., Josset, D., Ferlay, N., Peers, F., Thieuleux, F., Ducos, F., Pascal, N., Tanré, D., Pelon, J., and Goloub, P.: Consistency of aerosols above clouds characterization from A-Train active and passive measurements, Atmos. Meas. Tech., 10, 3499–3523, https://doi.org/10.5194/amt-10-3499-2017, 2017.

Chauvigné, A., Waquet, F., Auriol, F., Blarel, L., Delegove, C., Dubovik, O., Flamant, C., Gaetani, M., Goloub, P., Loisil, R., Mallet, M., Nicolas, J.-M., Parol, F., Peers, F., Torres, B., and Formenti, P.: Aerosol above-cloud direct radiative effect and properties in the Namibian region during the AErosol, RadiatiOn, and CLOuds in southern Africa (AEROCLO-sA) field campaign – Multi-Viewing, Multi-Channel, Multi-Polarization (3MI) airborne simulator and sun photometer measurements, Atmos. Chem. Phys., 21, 8233–8253, https://doi.org/10.5194/acp-21-8233-2021, 2021.

Line 57: "*Section 5 shows the data processing of one year (2008) PARASOL measurements and comparison with adjacent PARASOL-RemoTAP clear-sky aerosol retrievals.*"

A comparison with a similar algorithm would have been more relevant, given the inherent differences between aerosol concentrations integrated over the total atmospheric column (including low-altitude aerosols like marine aerosols) and those corresponding to aerosols above clouds.

Suggestion: The comparison between clear-sky and above-cloud aerosol retrievals could also focused on the fine mode Aerosol Optical Thickness (AOT). Such a comparison seems more relevant especially for biomass burning particles, which are predominantly fine mode and often found in elevated layers as for instance over the Southeast Atlantic region.

**2. Data Description / section 2.3**

Line 77: *"Here in this work, a pixel is marked as liquid phase only when the fraction of liquid-cloud-flagged 1-km-resolution MODIS pixels within a 6km × 6km PARASOL grid cell is larger than 80%."*

In Waquet et al. (2013), cloud optical thickness standard deviation was derived from 1-km-resolution MODIS retrievals within PARASOL pixels. They applied criteria to select only homogeneous POLDER pixels, based on spatial variability in cloud properties.

This allows to reduce the plan parallel effects that impact the modeling of polarize radiance especially in the cloud bow region (Cornet et al., 2013). This effect may result in false detection of aerosol above clouds (positive bias in the ACAOT)

Does your method control for sub-pixel cloud property heterogeneity by rejecting the most heterogeneous pixels? or is this neglected? Please clarify this point.

Please add Cornet et al., 2013 in the list of reference.

Cornet, Celine & C.-Labonnote, Laurent & Szczap, F. & Deaconu, Lucia-Timea & Waquet, Fabien & Parol, Frederic & Vanbauce, Claudine & Thieuleux, François & Riedi, J.. (2017). Cloud heterogeneity effects on cloud and aerosol above cloud properties retrieved from simulated total and polarized reflectances. Atmospheric Measurement Techniques Discussions. 1-25. 10.5194/amt-2017-413.

At the very last, mention the inherent limitations of using plane-parallel radiative transfer code for aerosol remote sensing in cloudy scenes

Line 110: "Only the measurements with a minimum of 14 angles are considered for the NN training, in order to evade from a variable-sized input vector to the NN or, as an alternative, an input vector with missing data."

This sentence is not unclear to me. Could you rephrase it or provide more explanation?

**Section 3.2: Neural network training.**

Line 159: *"To increase numerical efficiency and reduce memory usage during the training process, we choose the "neural network ensemble" approach (Hansen and Salamon, 1990)"*

Why did you choose the neural network ensemble? It typically requires significant data, computational power, and memory, which appears to contradict your goal of *"increasing numerical efficiency and reducing memory."*

Also, the reference Hansen and Salamon (1990) is quite old. Are there any more recent references on neural network ensembles?

How do you justify the use of an ensemble approach compared to using a classical method?

Please correctly write out the three proposed architectures:

- Show diagrams of the architectures.
- Present the hyperparameters for each architecture.

Describe the dataset for each step: what is used as input, the validation/test split, and include a table summarizing this information.

Also, it would be interesting to see the training curves for both validation and learning, so we can see the performance of your NNs

Line 169: *"The Adam optimizer (Kingma and Ba, 2014) is used to minimize the mean root square error (RMSE) loss function."*

Could you please specify the settings used for the Adam optimizer?

Line 123: *"In the training set, 20% of the samples represent the situation where the aerosol layer is located above the cloud top, in order to improve NN's ability to produce liquid and ice cloud fractions in areas of interest for this study. A pixel will be further processed"*

8 million data points, of which only 20% met the conditions. Why not use the correct number of data points directly if you're going to reduce it afterwards?

Line 121: *"with more cloud fractions close to 1 in order to acquire better sensitivity at almost fully cloudy cases"*

"Does this limit your reliable retrievals to areas with 100% cloud coverage? If so, please mention it. It would be useful to summarize the limitation(s) of your method in the conclusion section and abstract.

You mention that your state vector includes the cloud top altitude. Is this actually retrieved with your method? Have you compared your cloud top height retrievals with concomitant CALIOP data? If so, what is the robustness of your retrieval? What are the assumed aerosol base and top altitudes in your RT code?

Line 115: *"The first NN (liquid cloud mask) takes intensity, degree of linear polarization (DoLP), and viewing geometries (SZA, VZA, RAA and scattering angle) as input and outputs liquid cloud fraction and ice cloud fraction separately"*

The name of your first neural network, "liquid cloud mask," is a bit confusing. Since you're using it to estimate both liquid cloud fraction and cirrus cloud fraction, it seems to do more than a simple liquid cloud mask.

Also, how is your mask performing?

Line 143: *"The intensity and DoLP, as a function of wavelength and viewing angle, are compressed using a principal component analysis (PCA) before the training. A total of 25 principal components are retained for radiance and 33 for DoLP."*

Is the use of PCA indispensable? Please justify its inclusion, as its benefit is not immediately apparent when combined with a deep neural network.

Line 156: *"It should be noted that the NN forward model is not a complete forward model. It only works for pixels fully covered by a liquid cloud without any radiative contribution from the surface and is designed only for the purpose of goodness-of-fit assessment for above cloud aerosol retrievals."*

I'm not convinced the third network is truly necessary. Is it sufficiently accurate for predicting both total radiances and polarized radiances? How is its performance evaluated? It might be discarding valid retrievals if this NN is not accurate enough.

Line 161: *"The final output is the average of the outputs from all the ensembles"*

For the second NN, what are the discrepancies between the 16 networks? Are these discrepancies significant?

Line 169: *"and batch training with a batch size of 12,000"*

As the first reviewer noted, this value seems unusually high compared to what's reported in the literature. Please clarify.

**Section 4: synthetics measurements**

Figure 2 lack sufficient detail to evaluate the method's performance. Could you provide more metrics?

For instance, can you add linear fit results on the curves in Figures 2? and the number of considered points? it will be helpful.

For the results shown in Figure 2: Both absolute and relative Mean Absolute Errors (MAEs) should be provided. The results should be presented in tables.

Figure 2-e and Figure 2-h show the results with synthetic retrievals for the Ängström Exponent (AE).
I am surprised to see that the AE is systematically low biased for fine mode aerosols and high biased for coarse dust aerosol and the correlation coefficients are very low (<0.3).

I would expect to see random results scattered around the one-to-one line, similar to the general test results shown in Figure 2b.

Does this imply that your architecture is not adequately dimensioned to retrieve AE for extreme size distributions (e.g., purely fine or coarse modes)? If so, should the training be enhanced for

these extreme scenarios? Such extreme conditions are particularly representative of satellite observations for aerosols located above clouds.

Line 185: *"For AE and SSA, an additional mask of retrieved ACAOT > 0.2 is applied."*
-Please specify the wavelength for the ACAOT considered here.

Line 194: *"The retrievals are always masked by a retrieved liquid cloud fraction larger than 0.8"*
Could you recall the spatial resolution of your cloud mask?

Line 195: Same comment, please add wavelength for the ACAOT

Line 202: *"Over ocean, we see an opposite effect (except for very small COT), because the contribution from the ocean is relatively small and a smaller COT would even enhance the relative contribution of the aerosol signal compared to the cloud signal."*

Did you account for the surface wind speed and sun-glint in your method?

**5.1 Comparison between PARASOL-NN above cloud aerosol retrievals and adjacent RemoTAP clear-sky aerosol retrievals**

Similar to Figure 2, Figure 4 would benefit from additional metrics to properly evaluate the comparison results.

As previously discussed, the RemoTAP clear-sky algorithm results are not directly comparable with the above cloud aerosol properties retrieved with the present. It would have been more interesting to compare with existing aerosol above clouds available products.

Line 207: *"the data are aggregated at the same 1◦ × 1◦ grid cell"*.

Could you also provide a comparison between clear-sky and above-clouds retrievals for a case study (e.g., a daily product for a portion of an orbit)? This is also important to show the spatial variability in the retrieved aerosol above clouds properties obtained with your method.

For Figure 5, please adjust the color scale for the ACAOT. It's currently difficult to discern differences for ACAOT values between 0 and 0.1 (most of the values …). A histogram of ACAOT would be also very useful.

In Figure 5: What is the wavelength for the ACAOT?

Line 233: There seems to be an error in the article citation.

Please cite the paper by Waquet et al. (2013b) that presents a geophysical analysis of the global aerosol properties above clouds using POLDER by season for 2008. This study is directly comparable to yours (see Figure 1 in Waquet et al., 2013b).

Waquet, F., F. Peers, F. Ducos, P. Goloub, S. Platnick, J. Riedi, D. Tanré, and F. Thieuleux (2013b), Global analysis of aerosol properties above clouds, Geophys. Res. Lett., 40, 5809–5814, doi:10.1002/2013GL057482.

To avoid confusion, please differentiate between the two Waquet et al., 2013 (a) (remote sensing method) and (b) (geophysical analysis) references

Waquet, F., Cornet, C., Deuzé, J.-L., Dubovik, O., Ducos, F., Goloub, P., Herman, M., Lapyonok, T., Labonnote, L. C., Riedi, J., Tanré, D., Thieuleux, F., and Vanbauce, C.: Retrieval of aerosol microphysical and optical properties above liquid clouds from POLDER/PARASOL polarization measurements, Atmospheric Measurement Techniques, 6, 991–1016, https://doi.org/10.5194/amt-6-991-2013, 2013a

From Lines 236 to 241: The comparison of your results with those of Waquet et al. (2013) is too succinct and qualitative. I would favor a more quantitative comparison, at least for some case studies.

Line 245: *"We have to remark that our AE in regions between $45° − 60°N$ and $45° − 60°S$ is $\sim$ 0.8, which differs largely from $\sim 1.8$ in Waquet et al. (2013), despite the good agreement of our above cloud AE with the adjacent clear-sky AE in these latitudes."*

This funding is interesting and deserves more investigation.

Please add this information in the manuscript: the above-clouds AOTs associated with an AE of 1.8 in Waquet et al. (2013a) method for the 45°−60°N region are typically low (<0.05 at 865 nm), and even lower for the 45°−60°S region (<0.03 at 865 nm)

My opinion is that the ACAOTs are probably too low for effective aerosol type identification.

-What are your ACAOT values for these cases (i.e., cases with an AE of about 0.8)? Please add the corresponding ACAOT map to Figure 6

- Line 245: our AE in regions between $45° − 60°N$ and $45° − 60°S$ is$\sim$ 0.8

What would be the source of these particles located above clouds?

For such retrieved AE values (AE of about 0.8), this means that your algorithm retrieves a mixture of non-spherical mineral dust and fine mode particles.

Is your clear-sky algorithm also detect non-spherical coarse mode (mineral dust) over these regions for adjacent cases?

What would be the source of these mineral dust particles located above clouds over the $45° − 60°S$ region in the south hemisphere?

**Additional comments:**

It would be useful to remind the reader of the operational retrieval strategy previously developed for POLDER. This will help them understand any potential differences between that

method and your current approach. Some of the following explanations should be incorporated into the manuscript.

**Initial Test Phase:** Initially, the algorithm focuses on fine mode aerosols (considering six distinct models) and utilizes only forward and side scattering angles (scattering angle less than 130°). This range is chosen because, for Cloud Optical Thickness (COT) greater than 3, the polarized radiance reflected by the cloud is low (or weakly negative) and stable, becoming independent of cloud microphysics and optical thickness. Therefore, any additional polarization observed within this scattering angle range indicates the presence of a lofted aerosol layer above the clouds. (For a schematic view, refer to Figure 4 in Waquet et al., 2013a).

**Full Retrieval Phase:** If the retrieved above-cloud Aerosol Optical Thickness (AOT) from this initial test exceeds 0.1 at 865 nm, the algorithm proceeds to search for the best-fitting model among *all* available models. This comprehensive set includes the six fine modes plus a bimodal non-spherical mineral dust particle model, and the retrieval uses all available data across all scattering angles.

Conversely, if the above-cloud AOT is less than or equal to 0.1 at 865 nm, the AERO-AC algorithm solely retrieves the fine mode AOT and fine mode Angstrom Exponent (AE) using only observations for scattering angle smaller than 130°. So, for these situations, the AE values are the constrained between 1.6 and 2.4. This previous restriction on using the cloud bow and dust model in the AERO-AC algorithm was implemented to prevent false detection of AC events above clouds, particularly when the above-cloud atmosphere appears pristine based on initial tests.

**Regional Limitations:** It's important to note that in the regions between 45°–60°N and 45°–60°S, the above-cloud AOTs retrieved by the AERO-AC algorithm are likely too low to facilitate accurate aerosol type identification. This is a probable explanation for the observed differences between the mean AERO-AC results and those derived using the present method.

---

## Author Comment (AC1)

**Response to reviewer 1**

We would like to thank the reviewer for his/her important comments and suggestions.

The manuscript presents a neural network-based approach to above-liquid-cloud aerosol retrievals from the multi-angle polarimetric measurements by PARASOL. The method utilizes 3 separate NNs: one to determine if there is a liquid cloud layer, one to perform the ACA retrieval, and one for an approximate forward model for goodness-of-fit evaluation. Seasonal distributions of above-liquid-cloud aerosol optical thickness appear to generally agree with past studies. However, the manuscript lacks crucial details and tests that preclude its publication in the present state. Detailed comments are provided below.

☐ 0. A high-level comment: there are some acronyms that are defined in multiple places in the manuscript (for example, see comment #3), as well as places where the defined acronyms are not used, e.g., lines 14-15 spell out "above cloud aerosol" and "direct radiative effect" rather than using the ACA and DRE acronyms that were defined in the sentences before it. It would be good to define these acronyms only at their first usage and consistently use them throughout the manuscript.
We have a thorough check among the whole text and revised them

☐ 1. Line 7: "ACAOT (above cloud aerosol optical depth)" should be changed for consistency, either to "ACAOT (above cloud aerosol optical thickness)" or "ACAOD (above cloud aerosol optical depth)".
We changed ACAOD to ACAOT and use it consistently throughout the revised manuscript

☐ 2. Line 64-65: "The level 1 data are provided on a common sinusoidally grid of approximately with ground pixels of approximately 6 × 6 km2." - This sentence phrasing is confusing and I am not entirely sure of its intended meaning. Please rephrase it to more clearly convey the intended meaning.
We rephrased it as "The level 1 data are provided on a 6x6 km2 sinusoidally grid."(line 69 of the revised paper)

☐ 3. Line 89: The AOT acronym was defined earlier in the text on line 45, it doesn't need to be re-defined here.
We removed this definition.

4. Section 3.2: While this section does a great job explaining how the training data were produced, it lacks sufficient detail for other aspects relevant to NN training:

☐ a) Lines 154-155: Real measurements would have noises that vary across different measurements. Why is the noise assumed to be a constant relative noise of 0.02 for the intensity and a constant absolute noise of 0.012 for DoLP? These seem arbitrarily chosen, based on the provided information. This also suggests that the model will not be as accurate when noise levels deviate from these assumed values. Were other noise levels considered? See also comment #5d below.
This noise setting is only used in the calculation of chi2 (equation 2 in the preprint), but for the training, a variable intensity noise of 1-3% is considered while the noise of DoLP is constant at 0.012. We investigated different noise settings on intensity and DoLP (fixed value and variable range) in the training, both in this work and previous work (Yuan et al 2024) and the current setting produces the best results for cloud fraction retrievals (Yuan et al, 2024) as well as above-cloud aerosol retrieval (this paper), although the effect of the noise setting is minor. We would like to note that also the two main aerosol retrieval algorithms for PARASOL (GRASP and RemoTAP) assume a constant noise for evaluation of goodness-of-fit and during the inversion process. See Hasekamp et al. (2024) and references therein.

☐ b) Lines 162-163: How was the ensemble size for each model component determined?
This was empirically determined using synthetic experiments. We added this information to the revised manuscript (line 184)

☐ c) Lines 159-161: Regarding the description of NN ensembles, the current phrasing suggests that the described approach ("the whole training set is equally and randomly divided into several parts") is the only way to perform this, but in reality there are many other methods to achieve NN ensembles that do not follow this procedure (see, e.g., Dietterich, 2000). Please rephrase this sentence so that it is clear that this is your elected methodology to achieve an ensemble of NNs, not that this is the only methodology to achieve it.
We have rephrased it to "in our approach, the whole training set is equally and randomly divided into several parts (line 179 of the revised paper)

☐ d) Lines 165-166: For the cloud mask and retrieval NN, how was the measurement noise model determined? These values seem arbitrarily chosen based on the provided information.

The noise setting is inherited from Yuan et al 2024, where different noise settings are investigated for cloud mask from PARASOL measurements (see also our response above).

☐ e) Line 169: How was the batch size of 12,000 selected for this problem, and were other batch sizes considered? This is an unusual value, as typical batch sizes are chosen as 2 to some power, for efficiency when using a GPU (see, e.g., Kandel & Castelli, 2020). This is also unusual given the magnitude, as batch sizes are often significantly smaller than this; existing literature suggests that small batch sizes perform better (e.g., Bengio, 2012; Masters & Luschi, 2018; Kandel & Castelli, 2020).

Indeed, a smaller batch size can help in increasing the generalization and decreasing the memory used in the training process. Nevertheless, a large batch size benefits the convergence rate (Soham De, et al, 2017). Compared to applications such as image processing, one training sample in this paper consumes less memory, which makes it possible to use a larger batch size. We did several tests for different batch sizes (from 512 to 20000) and didn't find significant differences over the NN's performance. We added a statement on it in the revised paper (line 190)

☐ f) Line 170: "mean root square error (RMSE)" should be fixed to "root mean square error (RMSE)".

Corrected.

☐ g) Lines 170-172: How were the model architectures determined? It is surprising that a given model has the same number of neurons in each layer, and that all 3 models have the same number of hidden layers.

1) We don't find an obvious increase of performance from using different number of neurons in each layer in this application. However, using different number of neurons in different layers will lead to a non-square weight matrix, which has a potential risk of rank diminishing in forward and backward propagation. 2) In all the NN retrieval experiments (from PARASOL measurements), we found a worse performance (from the holdout set) when the number of layers increased or decreased.

☐ h) What activation functions were used, and how were they determined?

We use ReLU as activation function. but we have tried different activation functions (ReLU, leaky ReLU, sigmoid, tanh, etc) and no obvious performance difference is observed. Therefore, for computational efficiency, we chose the "simplest" ReLU as activation function. We mentioned this in the revised paper (line 193)

☐ i) What learning rates were used to train these models? How were those learning rates determined?

Initial learning rate is 0.01 which is default and suggested for the Adam optimizer. The Adam optimizer changes the learning rate based on the convergence situation at each

iteration step and is not very sensitive to the initial value of the learning rate. We added a description on it in the revised manuscript.

☐ j) Line 119: How were the 8 million samples split into subsets for training, validation, and testing?
Ninety percent of the samples are in the training set and the rest are in the holdout (test) set. The holdout set is only used for assessing the training process. The validation/evaluation of the NN approach is based on an individual data set that is independent from the training procedure. We added the information to the revised script.(line 180)

☐ k) Line 144: How did you determine the number of leading PCs to use for each variable? How much explained variance do these PCs comprise? Did you first attempt this without using PCA and find poor results?
We tried different values to find the best settings (from the performance of synthetic experiments), and we also tried without PCA, which performs not as good as the current setting. For intensity, 33 PCs comprise 99.99% explained variance. For DoLP, 25 PCs comprise 99.14% explained variance. We included this in the revised paper (line 157).

5. Section 4.1 is missing some key synthetic tests/results:

☐ a) What is the confusion matrix for the NN liquid cloud mask model? This is a crucial table to provide for any classification NN to better understand the rate of true/false positives/negatives and thereby the reliability of the NN for this task.
We added a confusion matrix (as shown below) to the supporting information:

[Figure]

Fig 1. Confusion matrix of liquid cloud mask, left is over ocean and right is over land.

☐ b) What is the accuracy of the goodness-of-fit determination by the NN forward model? For a given state vector retrieved by the NN retrieval model, a forward model can be computed by either the NN forward model or the physics-based forward model. Doing

both for all cases considered in this synthetic test will enable the determination of the reliability of the NN forward model for this task based on metrics like the correlation coefficient, coefficient of determination, RMSE, etc.

Below we show the comparison of intensity and degree of linear polarization (DoLP) between NN forward model and RemoTAP forward model, at 565nm. The rstd (relative standard deviation) of intensity is 0.7% and the std (standard deviation) of DoLP is 0.0025, both of which are below the instrument measurement noise. This suggests that the NN forward model is good enough to replace the full physical model (RemoTAP) in estimation goodness-of-fit. We added the figures in the revised manuscript (Figure 2 of the revised manuscript).

[Figure]

Fig 2. Intensity (left) and degree of linear polarization (DoLP, right) from NN forward model (prediction) and RemoTAP forward model (truth) at 565nm.

☐ c) The correlations (I assume the R correlation coefficient? Please be explicit) reported here suggest that the NN model poorly captures AE behavior, especially under fine- or dust-dominated conditions. This is also the case for SSA under dust-dominated conditions. This seems to suggest that the retrieval method is only weakly sensitive to these parameters. How does this compare with physics-based retrievals of these parameters under these conditions? If it similarly struggles, it would be good to discuss this and clearly point out that this is not a limitation of the NN approach. However, if the physics-based retrievals do not have this issue, then it suggests that the presented NN-based approach is not optimal.

R indeed indicates the correlation coefficient. We have better clarified that in the revised manuscript. The relatively low correlation coefficients for the fine- and dust-mode dominated cases can largely be explained by the small range in AE, where most values are between 1.5-2,0 for for fine- dominated cases and between 0-0.5 for dust-dominated cases, where this range is close to the retrieval accuracy itself (RMSE of ~0.4). So, it means little capability to distinguish size within these categories, but clear capability to distinguish between fine-mode dominated and dust-mode dominated

cases (as shown in the top panel). A similar explanation holds for the dust SSA. Compared to clear-sky full-physics retrievals (Hasekamp et al, 2024), the performance for SSA of our above-cloud retrievals is similar. The performance for AE is worse than for clear-sky full physics retrievals, but this is most likely because of smaller information content for AE for above-cloud aerosol retrievals than for clear sky retrievals. Namely, performing clear sky aerosol retrievals with an NN gives similar results to full physics (this is current research ongoing in our group).

☐ d) Related to comment #4a: How does the model perform when applied to synthetic data with a different noise level than that assumed in Sec. 3.2? It would be helpful to understand how this impacts the results, given that real measurements will not exactly follow the noise model assumed when generating the training data.

Below we show three different noise level on the synthetic dataset with both fine and dust mode aerosols. It can be seen that increasing the measurements noise leads to the increase of retrieval error on all the three properties, but the change is small in this noise range (intensity noise 1-3% and DoLP 0.007-0.017). We include those figures in the supporting information (Fig 6 of SI).

[Figure]

[Figure]

Fig 3. ACAOT 550nm, AE 440-670nm and SSA 550nm on the same synthetic dataset with different noise levels. The first line's noise settings are 1% to intensity and 0.007 to DoLP. The second line is 2% to intensity and 0.012 to DoLP. The third line is 3% to intensity and 0.017 to DoLP.

6. Section 4.2 / Figure 3:

☐ a) For Fig 3's a-c panels, it looks as if only a single retrieval was performed at the 4 chosen optical depths. I assume this is really showing the **mean** RMSE over all 10,000 retrievals at a given optical depth, rather than the RMSE of a single retrieval as the labeling and caption currently indicates? For clarity it would be good to include error bars showing the standard error of the mean, which will better show whether the observed trends are statistically significant. Please also update the caption to clarify that these RMSE values are averaged over the 10,000 cases considered at each COT value.

The RMSE is calculated over all the 10000 retrievals for a single wavelength (550 nm). Below shows the ACAOT 550, AE 440-670 and SSA 550 error (prediction - truth) as a function of COT, the error bars stand for the standard deviation of all the retrievals in a COT value. The figure indicates that the retrieval error on the dataset is dominated by the standard deviation. We included this information in the caption of Figure 4 in the revised manuscript, but prefer to keep the plot with RMSE instead of the plot below.

[Figure]

Fig 4. Error and standard deviation of ACAOT, AE and SSA retrievals in different COT bins. Blue lines are over ocean and orange lines are over land.

☐ b) Am I correctly inferring that the "number of remaining pixels" means the number of cases that were not screened out by the retrieved cloud fractions or goodness-of-fit

metric?  Assuming this is the case, rather than reporting the absolute number of pixels in Fig 3's d-f panels, it would be clearer to report the fraction of pixels where retrievals weren't screened out or, conversely, the fraction of pixels that were screened out.

We changed it to "fraction of successful retrievals" which is the fraction of pixels not screened out.

☐ 7. Lines 219-221: I don't necessarily agree with this statement.  Looking at SSA, the synthetic test found a correlation of ~0.76, with the correlation lowest for the dust-dominated cases at ~0.56.  Meanwhile, in Fig 4, the SSA correlation is even lower at ~0.37.  Fig 4 also shows a >27% increase in RMSD for SSA when compared to the RMSE of the synthetic tests.  From this, I would conclude that the AE is more comparable for above-cloud and adjacent clear-sky retrievals than AOT or SSA.

Indeed the AE looks most comparable between ACA retrievals and clear-sky retrievals among the three properties (ACAOT, AE and SSA). We rephrased it to "the intrinsic aerosol properties (AE and SSA) are more comparable for above-cloud and adjacent clear-sky retrievals than the AOT, although the correlation of SSA is low (0.37)" (line 250)

☐ 8. Line 245: "The high AE and low SSA is an expected feature of the smoke in mid-Africa."  Can you please provide a reference for this statement?

Mallet et al 2024 (https://doi.org/10.5194/acp-24-12509-2024) clearly stated this feature. We included this in the revised manuscript. (line 304)

☐ 9. Lines 245-247: Such a large disagreement in AE suggests some fundamental or systematic difference between the method considered here vs. that considered by Waquet et al. (2013), and it would be good to understand the origin of this difference.  Is there a reasonable explanation why the reported AEs are less than 1/2 that reported by Waquet et al. in these midlatitudes?  Does Waquet et al. also only consider above-liquid-cloud AE?  If not, given that above-ice-cloud AE is disregarded here, is that perhaps a common occurrence and the results of  Waquet et al. are elevated due to that above-ice-cloud AE?  Does the Waquet et al. approach overestimate ACAOT, or does the RemoTAP method considered here underestimate AE?  Did Waquet et al. similarly consider the differences between above-cloud AE with adjacent clear-sky AE?  Hopefully these questions can help to better elucidate the origin of this stark difference.

In the revised manuscript, we performed a more detailed comparison with AERO-AC (setction 5.2, see also our response to reviewer-2)

☐ 10. Lines 255-257: I don't agree that the synthetic experiments indicate the NNs have the ability to retrieve AE from fine- and dust-mode-dominated aerosol. Figure 3 shows that the performance is poor in these regimes: the NN underestimates the AE for fine-dominated cases, and it overestimates the AE for dust-dominated cases.

See our response above. We rephrased it to: ... the NNs have to ability to retrieve AE that allows separation between fine-mode and dust dominated cases ..." (line 319)

11. Lines 273-275: "... the NN-based surrogate forward model, just like the full-physical model, can provide goodness-of-fit mask to filter unphysical retrievals, which may due to imperfect cloud mask or some challenging aerosol/cloud/surface combination.":

☐ a) I don't see where this is substantiated in the manuscript; no comparisons are performed between the NN's goodness-of-fit calculation and a physics-based model's corresponding goodness-of-fit metric. Please perform the test in comment #5b to substantiate this claim.

We added the evaluation of NN forward model to the revised manuscript (see above).

☐ b) The last clause of this sentence is phrased awkwardly, and I cannot discern the intended meaning in the context of the rest of the sentence. Please rephrase this so that it clearly conveys the intended meaning.

We rephrased it as "... unphysical retrievals. Those unphysical retrievals are caused by reasons such as imperfect cloud screening or challenging aerosol/cloud/surface combinations" (line 342)

---

## Author Comment (AC2)

**Response to reviewer 2**

We would like to thank the reviewer for his/her important comments and suggestions.

*Abstract.*

☐ Please add the wavelength(s) used for ACAOT, AE and SSA

The wavelengths are added in the revised manuscript.

*Introduction.*

So, this section requires improved referencing and description of the operational aerosol above cloud algorithm previously developed for POLDER/PARASOL, including the associated available product and validation efforts. The AERO-AC product, with its DOI, is globally available for 5 years of POLDER data, which is worth noting for the reader.

☐ Some of the following explanations should be incorporated into the manuscript (see also my additional comments at the end of this review).

Initially, the Waquet et al. (2009, 2013) method determined above-cloud aerosol optical thickness and Ångström exponent exclusively from polarization measurements. This was achieved using a look-up table (LUT) approach combined with a decision tree strategy.

-The method was then improved by including additional total radiance measurements (Peers et al., 2015) to simultaneously retrieve the above cloud aerosol single scattering albedo and the cloud optical thickness of the below cloud layer (COT).

-The associated global product is referred to as AERO-AC (Waquet et al., 2020)

The aerosol above cloud properties are only retrieved in case of homogeneous optically thick (COT > 3) and liquid water clouds. Cloud fractional covers and cloud edges are removed. Cirrus above liquid water clouds are also filtered and different quality criteria are eventually applied to improve the products.

In the revised manuscript, we added the suggested references and the related explanations to the introduction (line 40) and data description (line 85)

☐ Please add the following references:

Peers, F., Waquet, F., Cornet, C., Dubuisson, P., Ducos, F., Goloub, P., Szczap, F., Tanré, D., and Thieuleux, F.: Absorption of aerosols above clouds from POLDER/PARASOL measurements and estimation of their direct radiative effect, Atmos. Chem. Phys., 15, 4179–4196, https://doi.org/10.5194/acp-15-4179-2015, 2015.

Waquet F., Peers F., Ducos F., Thieuleux F., Deaconu L., A. Chauvigné and Riedi, J.: Aerosols above clouds products from POLDER/PARASOL satellite observations (AERO-AC products), doi:10.25326/82, 2020.

The references are included (line 42 and 85).

☐ Please mention the methods that use active measurements to retrieve aerosol properties above clouds. Different methods (standard methods and advanced methods like the "depolarization ratio method") were developed for CALIOP and various products are available (see Jethva et al., (2014) and Deaconu et al. (2017))

Those methods are included in the paper (line 45).

☐ It's also important to highlight the research community's dedication to validating and intercomparing their passive and active aerosol-above-cloud products. This has involved rigorous work, ranging from in-depth case study analyses (Jethva et al., 2014) — supported by airborne sun-photometer data (Chauvigné et al., 2021) — to comprehensive global scale analyses (Deaconu etal., 2017).

Please add the following references:

Jethva, H., O. Torres, F. Waquet, D. Chand, and Y. Hu (2014), How do A-train sensorsintercompare in the retrieval of above-cloud aerosol optical depth? A case study-based assessment, Geophys. Res. Lett., 41, 186–192, doi:10.1002/2013GL058405.

Deaconu, L. T., Waquet, F., Josset, D., Ferlay, N., Peers, F., Thieuleux, F., Ducos, F., Pascal, N., Tanré, D., Pelon, J., and Goloub, P.: Consistency of aerosols above clouds characterization from A-Train active and passive measurements, Atmos. Meas. Tech., 10, 3499–3523, https://doi.org/10.5194/amt-10-3499-2017, 2017.

Chauvigné, A., Waquet, F., Auriol, F., Blarel, L., Delegove, C., Dubovik, O., Flamant, C., Gaetani, M., Goloub, P., Loisil, R., Mallet, M., Nicolas, J.-M., Parol, F., Peers, F., Torres, B., and Formenti, P.: Aerosol above-cloud direct radiative effect and properties in the Namibian

region during the AErosol, RadiatiOn, and CLOuds in southern Africa (AEROCLO-sA) field campaign – Multi-Viewing, Multi-Channel, Multi-Polarization (3MI) airborne simulator and sun photometer measurements, Atmos. Chem. Phys., 21, 8233–8253, https://doi.org/10.5194/acp-21-8233-2021, 2021.

The references above are included and the work from the community is highlighted in the revised manuscript.(line 52 and 232)

Line 57: "Section 5 shows the data processing of one year (2008) PARASOL measurements and comparison with adjacent PARASOL-RemoTAP clear-sky aerosol retrievals."

A comparison with a similar algorithm would have been more relevant, given the inherent differences between aerosol concentrations integrated over the total atmospheric column (including low-altitude aerosols like marine aerosols) and those corresponding to aerosols above clouds.

☐ Suggestion: The comparison between clear-sky and above-cloud aerosol retrievals could also focused on the fine mode Aerosol Optical Thickness (AOT). Such a comparison seems more relevant especially for biomass burning particles, which are predominantly fine mode and often found in elevated layers as for instance over the Southeast Atlantic region.

We have added a subsection (5.2, to the paper with a comparison with the AERO-AC product to the paper. We believe both comparisons (i.e. to nearby clear-sky retrievals and AERO-AC) retrievals have their own specific relevance. For example, we expect that ACAOT is generally correlated with total AOT, but ACAOT should be smaller. For situations with larger AOD (>0.2), the intrinsic aerosol properties are expected to have many similarities between above-cloud cases and clear-sky cases, and plausible explanations can be found for remaining differences (e.g. at high AE we expect the above-cloud AE to be slightly larger than total column AE, because it is less influenced by Sea Salt).

We have also. added the fine mode AOT comparison between clear-sky and above-cloud aerosol retrievals (Figure 6 in the revised manuscript).

*2. Data Description / section 2.3*

Line 77: "Here in this work, a pixel is marked as liquid phase only when the fraction of liquid-cloud-flagged 1-km-resolution MODIS pixels within a 6km × 6km PARASOL grid cell is larger than 80%."

In Waquet et al. (2013), cloud optical thickness standard deviation was derived from 1-km-resolution MODIS retrievals within PARASOL pixels. They applied criteria to select only homogeneous POLDER pixels, based on spatial variability in cloud properties.

This allows to reduce the plan parallel effects that impact the modeling of polarize radiance especially in the cloud bow region (Cornet et al., 2013). This effect may result in false detection of aerosol above clouds (positive bias in the ACAOT)

☐ Does your method control for sub-pixel cloud property heterogeneity by rejecting the most heterogeneous pixels? or is this neglected? Please clarify this point.

Please add Cornet et al., 2013 in the list of reference.

Cornet, Celine & C.-Labonnote, Laurent & Szczap, F. & Deaconu, Lucia-Timea & Waquet, Fabien & Parol, Frederic & Vanbauce, Claudine & Thieuleux, François & Riedi, J.. (2017). Cloud heterogeneity effects on cloud and aerosol above cloud properties retrieved from simulated total and polarized reflectances. Atmospheric Measurement Techniques Discussions. 1-25. 10.5194/amt-2017-413.

In our method, there is no such control, but we expect that the goodness-of-fit criterion filters out many of these situations, because they will cause variations between viewing angles that cannot be modeled by the 1D forward model (Stap et al., 2015; 2016). We added a discussion on this topic to the revised manuscript and included the reference to Cornet et al. (section 3.2, line 126).

☐ At the very last, mention the inherent limitations of using plane-parallel radiative transfer code for aerosol remote sensing in cloudy scenes

Now the limitations are mentioned in the paper.(section 3.2, line 126)

☐ Line 110: "Only the measurements with a minimum of 14 angles are considered for the NN training, in order to evade from a variable-sized input vector to the NN or, as an alternative, an input vector with missing data."

This sentence is not unclear to me. Could you rephrase it or provide more explanation?

PARASOL-POLDER can observe a ground pixel at up to 16 angles, but the number of viewing angles varies over the different L1C pixels. The majority of pixels observers a ground pixel at 14 angles and that is what we trained our NN for. To train an NN for a variable size of the input

vector is very challenging. In principle a better approach would be to train separate NNs for different sizes of the input. We clarified that in section 2.1, line 70.

[Figure]

Fig 1. Histogram of PARASOL available viewing angles per pixel.

*Section 3.2: Neural network training.*

☐ Line 159: "To increase numerical efficiency and reduce memory usage during the training process, we choose the "neural network ensemble" approach (Hansen and Salamon, 1990)"

Why did you choose the neural network ensemble? It typically requires significant data, computational power, and memory, which appears to contradict your goal of "increasing numerical efficiency and reducing memory."

Using "neural network ensemble" approach can significantly reduce the NN's overfitting and increase NN's generalization (Ortega et al, 2021). Based on our experiments (both in this paper and previous studies), a similar performance can be achieved by training all samples in one go or separating them into several ensembles (e.g., an NN trained with 16 million samples or 16 NNs trained with 1 million samples each). However, the latter (NN ensemble method) requires far less total training time and memory, and different ensembles can be trained simultaneously on different computing nodes individually. Based on the mentioned merits, we chose NN ensemble approach. It is true that when applying the NN, an ensemble approach has a higher computational cost, but still this is negligible compared to full physics algorithms.

☐ Also, the reference Hansen and Salamon (1990) is quite old. Are there any more recent references on neural network ensembles?

M.A. Ganaie, et al, (2022) wrote a review of the development of neural network ensemble strategies, including badging, boosting, stacking, etc (mainly on classification application). In our paper, we use the approach in Hansen and Salamon (1990) (therein they use the majority voting scheme for classification, while we use the averaging strategy). We add Ganaie, et al (2022) as reference as well in the revised manuscript (line 179).

☐ How do you justify the use of an ensemble approach compared to using a classical method?

Please see my answer above about the justification for the ensemble approach.

☐ Please correctly write out the three proposed architectures:

Show diagrams of the architectures.
Present the hyperparameters for each architecture.

Describe the dataset for each step: what is used as input, the validation/test split, and include a table summarizing this information.

Diagram (Fig 1 in SI, NN ensemble structure) and table (Table 1 in SI, three NNs' details, e.g, input, output, etc) are added to show the NN architectures as well as the inputs and outputs of the different NNs in the SI of the revised manuscript.

☐ Also, it would be interesting to see the training curves for both validation and learning, so we can see the performance of your NNs

Here we show the loss function of training set and holdout set from one ensemble in the ACA retrieval NN, and it can be seen that the loss function on the two sets both converges well without overfitting features. We added these figures to the SI of the paper (Fig 2 in SI).

[Figure]

Fig 2. Loss function v.s training epoch on training set (left) and holdout set (right).

☐ Line 169: "The Adam optimizer (Kingma and Ba, 2014) is used to minimize the mean root square error (RMSE) loss function."

Could you please specify the settings used for the Adam optimizer?

The optimizer settings include: learning rate = 0.001, betas = (0.9, 0.999), eps = 1e-8 and weight_decay = 0. The settings are the default recommended settings. The adam optimizer is an adaptive optimizer and based on our experience, it is virtually not sensitive to the initial learning rate. We included this information in the revised manuscript (line 192).

☐ Line 123: "In the training set, 20% of the samples represent the situation where the aerosol layer is located above the cloud top, in order to improve NN's ability to produce liquid and ice cloud fractions in areas of interest for this study. A pixel will be further processed"

8 million data points, of which only 20% met the conditions. Why not use the correct number of data points directly if you're going to reduce it afterwards?
This line describes the training set of cloud mask NN. For the aerosol retrieval NN, all pixels in the training set are with aerosol above clouds. We have clarified this in the revised manuscript (line 160)

☐ Line 121: "with more cloud fractions close to 1 in order to acquire better sensitivity at almost fully cloudy cases"

"Does this limit your reliable retrievals to areas with 100% cloud coverage? If so, please mention it. It would be useful to summarize the limitation(s) of your method in the conclusion section and abstract.

This does limit reliable retrievals to large cloud fractions (CF > 0.80) but not just fully cloudy pixels. We clarified this in the conclusion of the paper (line 137).

☐ You mention that your state vector includes the cloud top altitude. Is this actually retrieved with your method? Have you compared your cloud top height retrievals with concomitant CALIOP data? If so, what is the robustness of your retrieval? What are the assumed aerosol base and top altitudes in your RT code?
Yes, the ACA retrieval NN outputs the full state vector including cloud top height (CTH), but from the performance over holdout set (test set), the CTH is not well retrieved (correlation is 0.56 and RMSE is 600 (m)). Therefore, we didn't compare it with CALIOP data. The aerosol profile follows a Gaussian distribution with a fixed FWHM=2000 m, and we retrieve only the center altitude (aerosol layer height). We clarified it in the revised manuscript (line 103)

□ Line 115: "The first NN (liquid cloud mask) takes intensity, degree of linear polarization (DoLP), and viewing geometries (SZA, VZA, RAA and scattering angle) as input and outputs liquid cloud fraction and ice cloud fraction separately"

The name of your first neural network, "liquid cloud mask," is a bit confusing. Since you're using it to estimate both liquid cloud fraction and cirrus cloud fraction, it seems to do more than a simple liquid cloud mask. Also, how is your mask performing?
We changed the name "liquid cloud mask NN" to "cloud mask NN". The performance of the mask is shown in the figures below. The figures are also included in the SI of the revised article (Fig 3 of SI)

[Figure]

Fig 3. Confusion matrix of liquid cloud detection on the holdout set, "pred 1" means predicted liquid cloud fraction > 0.8, "true 1" means true liquid cloud fraction > 0.8.

□ Line 143: "The intensity and DoLP, as a function of wavelength and viewing angle, are compressed using a principal component analysis (PCA) before the training. A total of 25 principal components are retained for radiance and 33 for DoLP."

Is the use of PCA indispensable? Please justify its inclusion, as its benefit is not immediately apparent when combined with a deep neural network.

An acceptable result can be obtained without PCA, but using PCA makes the results slightly better (from synthetic test) as a way of denoising.

□ Line 156: "It should be noted that the NN forward model is not a complete forward model. It only works for pixels fully covered by a liquid cloud without any radiative contribution from the surface and is designed only for the purpose of goodness-of-fit assessment for above cloud aerosol retrievals."

I'm not convinced the third network is truly necessary. Is it sufficiently accurate for predicting both total radiances and polarized radiances? How is its performance evaluated? It might be discarding valid retrievals if this NN is not accurate enough.

Below we show the comparison of intensity and degree of linear polarization (DoLP) between NN forward model and RemoTAP forward model, at 565nm. The rstd (relative standard deviation) of intensity is 0.7% and the std (standard deviation) of DoLP is 0.0025, both of which are below the instrument measurement noise. This suggests that the NN forward model is good enough to replace the full physical model (RemoTAP) in estimation goodness-of-fit. We added these figures to the revised manuscript (Figure 2 of the revised manuscript).

[Figure]

Fig 4. Intensity (left) and degree of linear polarization (DoLP, right) from NN forward model (prediction) and RemoTAP forward model (truth) at 565nm.

Additionally, the figures below show the comparison between RemoTAP clear sky retrieval and the NN ACA retrieval (as is in section 4) but without the goodness-of-fit chi2 mask derived from the 3rd network. It is clear to see the chi2 mask filtered out a lot of unphysical retrievals and improved the performance. The figures are included in the SI (Fig 4 of SI)

[Figure]

Fig 5. NN ACA retrievals v.s. adjacent PARASOL-RemoTAP clear sky retrievals. No goodness-of-fit mask applied. Other filters are the same as in section 4.1 of the paper.

☐ Line 161: "The final output is the average of the outputs from all the ensembles"

For the second NN, what are the discrepancies between the 16 networks? Are these discrepancies significant?

Below shows the ACAOT (550nm) mean retrieved value and the range across the different ensemble members from randomly chosen 100 pixels (1% of all) on the synthetic validation dataset used in the paper (both fine and dust mode aerosol). The average spread (max – min) is 0.067

[Figure]

Fig 6. ACAOT (550nm) spread among 100 pixels (1%) on the synthetic validation set.

Line 169: "and batch training with a batch size of 12,000"

As the first reviewer noted, this value seems unusually high compared to what's reported in the literature. Please clarify.

One motivation for the smaller batch size (compared to other works) is to decrease the memory used in the training process. However, a large batch size benefits the convergence rate (Soham De, et al, 2017). We did several tests over batch size (from 512 to 20000) and didn't find significant differences over the NN's performance for our application. We added a related statement in the revised manuscript (line 189).

*Section 4: synthetics measurements*

☐ Figure 2 lack sufficient detail to evaluate the method's performance. Could you provide more metrics? For instance, can you add linear fit results on the curves in Figures 2? and the number of considered points? it will be helpful.

We added more metrics in the plots and now it shows relative mean squared error (RMSE), mean absolute error (MAE), correlation coefficient (corr), number of pixels (npix) and coefficient of determination ($R^2$).

☐ For the results shown in Figure 2: Both absolute and relative Mean Absolute Errors (MAEs) should be provided. The results should be presented in tables.

We added MAE in the plots, but we believe relative MAE is not a good metric (especially for properties that can become close to zero) so it is not included. A table showing RMSE, MAE and bias of the synthetic experiments is included in SI (Table 2 of SI).

☐ Figure 2-e and Figure 2-h show the results with synthetic retrievals for the Ängström Exponent (AE). I am surprised to see that the AE is systematically low biased for fine mode aerosols and high biased for coarse dust aerosol and the correlation coefficients are very low (<0.3). I would expect to see random results scattered around the one-to-one line, similar to the general test results shown in Figure 2b.

Does this imply that your architecture is not adequately dimensioned to retrieve AE for extreme size distributions (e.g., purely fine or coarse modes)? If so, should the training be enhanced for these extreme scenarios? Such extreme conditions are particularly representative of satellite observations for aerosols located above clouds.

It is possible that the NN performance may be improved for these extreme scenarios by adding more of such samples in the training set. We added a discussion on this aspect in the revised manuscript. (line 207)

Line 185: "For AE and SSA, an additional mask of retrieved ACAOT > 0.2 is applied."
-Please specify the wavelength for the ACAOT considered here.

The ACAOT here means the ACAOT at 550nm, we clarified this (and also other ACAOT) in the revised manuscript.

☐ Line 194: "The retrievals are always masked by a retrieved liquid cloud fraction larger than 0.8" Could you recall the spatial resolution of your cloud mask?

The liquid cloud fraction is a direct output of the 1st NN, which is at the original PARASOL resolution (6 km x 6 km). We also add this into the revised paper (line 215).

☐ Line 195: Same comment, please add wavelength for the ACAOT

We added the wavelength to all the ACAOT in the revised paper.

☐ Line 202: "Over ocean, we see an opposite effect (except for very small COT), because the contribution from the ocean is relatively small and a smaller COT would even enhance the relative contribution of the aerosol signal compared to the cloud signal." Did you account for the surface wind speed and sun-glint in your method?

The wind speed variation have been taken into account in the training set. The geometry used to generate the training set is randomly taken from PARASOL real geometry (as is described in line 118 of the revised paper), which also include sun-glint areas.

*5.1 Comparison between PARASOL-NN above cloud aerosol retrievals and adjacent RemoTAP clear-sky aerosol retrievals*

☐ Similar to Figure 2, Figure 4 would benefit from additional metrics to properly evaluate the comparison results. As previously discussed, the RemoTAP clear-sky algorithm results are not directly comparable with the above cloud aerosol properties retrieved with the present. It would have been more interesting to compare with existing aerosol above clouds available products.
We have added additional metrics to the figures and a comparison with the PARASOL AERO-AC data product is included in the paper (see also our response above)

☐ Line 207: "the data are aggregated at the same 1◦ × 1◦ grid cell".

Could you also provide a comparison between clear-sky and above-clouds retrievals for a case study (e.g., a daily product for a portion of an orbit)? This is also important to show the spatial variability in the retrieved aerosol above clouds properties obtained with your method.
We have added a case study in mid-Africa on 04 Aug 2008, showing above-cloud and clear-sky retrievals (see below). They are included in the paper.

[Figure]

Fig 7. NN above cloud aerosol retrievals compared to RemoTAP clear sky aerosol retrievals in mid-Africa, 04 Aug 2008.

☐ For Figure 5, please adjust the color scale for the ACAOT. It's currently difficult to discern differences for ACAOT values between 0 and 0.1 (most of the values …). A histogram of ACAOT would be also very useful. In Figure 5: What is the wavelength for the ACAOT?
 We changed the colorbar of ACAOT plot to log-scale, and a histogram (as below) is included in the SI. The ACAOT is at 550nm.

[Figure]

Fig 8. Histogram of ACAOT (550nm) for the whole year 2008 PARASOL-NN ACA retrievals.

☐ Line 233: There seems to be an error in the article citation.

Please cite the paper by Waquet et al. (2013b) that presents a geophysical analysis of the global aerosol properties above clouds using POLDER by season for 2008. This study is directly comparable to yours (see Figure 1 in Waquet et al., 2013b).

Waquet, F., F. Peers, F. Ducos, P. Goloub, S. Platnick, J. Riedi, D. Tanré, and F. Thieuleux (2013b), Global analysis of aerosol properties above clouds, Geophys. Res. Lett., 40, 5809–5814, doi:10.1002/2013GL057482.

To avoid confusion, please differentiate between the two Waquet et al., 2013 (a) (remote sensing method) and (b) (geophysical analysis) references

Waquet, F., Cornet, C., Deuzé, J.-L., Dubovik, O., Ducos, F., Goloub, P., Herman, M., Lapyonok, T., Labonnote, L. C., Riedi, J., Tanré, D., Thieuleux, F., and Vanbauce, C.: Retrieval of aerosol microphysical and optical properties above liquid clouds from POLDER/PARASOL polarization measurements, Atmospheric Measurement Techniques, 6, 991–1016, https://doi.org/10.5194/amt-6-991-2013, 2013a
We revised the paper based on this comment.

☐ From Lines 236 to 241: The comparison of your results with those of Waquet et al. (2013) is too succinct and qualitative. I would favor a more quantitative comparison, at least for some case studies.
We included another section (section 5.2) showing the comparison between PARASOL AERO-AC data product and the NN retrievals.

☐ Line 245: "We have to remark that our AE in regions between 45°− 60°N and 45°− 60°S is∼ 0.8, which differs largely from∼ 1.8 in Waquet et al. (2013), despite the good agreement of our above cloud AE with the adjacent clear-sky AE in these latitudes."

This funding is interesting and deserves more investigation.

Please add this information in the manuscript: the above-clouds AOTs associated with an AE of 1.8 in Waquet et al. (2013a) method for the 45°−60°N region are typically low (<0.05 at 865 nm), and even lower for the 45°−60°S region (<0.03 at 865 nm)

My opinion is that the ACAOTs are probably too low for effective aerosol type identification.

-What are your ACAOT values for these cases (i.e., cases with an AE of about 0.8)? Please add the corresponding ACAOT map to Figure 6
We only select where ACAOT_550 nm > 0.2 for plotting (and evaluating) AE of our PARASOL-NN The difference may be partly caused by low ACAOT cases in AERO-AC, but also in the direct comparison (including only larger ACAOT) we see much larger AE in AERO-AC. We added a discussion (section 5.2) of the revised paper.

☐ - Line 245: our AE in regions between 45°− 60°N and 45°− 60°S is∼ 0.8 What would be the source of these particles located above clouds? For such retrieved AE values (AE of about 0.8), this means that your algorithm retrieves a mixture of non-spherical mineral dust and

fine mode particles. Is your clear-sky algorithm also detect non-spherical coarse mode (mineral dust) over these regions for adjacent cases?

Yes, the clear-sky retrievals retrieve a small contribution from the dust mode here, but overall the coarse aerosols are dominated by Sea Salt here. Both RemoTAP and GRASP give a low (clear-sky) AE in this region (Figure 10 in Hasekamp et al, 2024). The AE from that paper and this ACA paper are all filtered by AOT 550nm > 0.2.

☐ What would be the source of these mineral dust particles located above clouds over the 45°−60°S region in the south hemisphere?
It is likely that the algorithm retrieves a contribution from the dust mode in the presence of coarse sea salt. But also dust may be present from e.g. Patagonia or Australia.

---

## Editor Decision (ED1)

The above-cloud optical depth (ACAOD) retrieval from observations by Ozone Monitoring Instrument (OMI) on the Aura satellite does not use radiance measurements at 470 and 860 nm as mistakenly implied by the authors. OMI's hyper-spectral coverage does not go beyond 500 nm. The OMI ACAOD retrieval technique uses observations at 354 and 388 nm to simultaneously retrieve above-cloud aerosol optical depth and absorption corrected cloud optical depth. The OMI near-UV application demonstrated for the first time the ACAOD retrieval capability based on satellite radiance measurements.

---

## Author Response (AR2)

**Response to the editor**

We would like to thank the editor for his important comments and suggestions.

• We revised the description of the "color ratio" method (line 50).